# Self-rectifying resistive memory in passive crossbar arrays

Kanghyeok Jeon[1,2,4], Jeeson Kim[2,4], Jin Joo Ryu[1], Seung-Jong Yoo[1,2], Choongseok Song[2], Min Kyu Yang[3], Doo Seok Jeong [2✉] & Gun Hwan Kim [1✉]

Conventional computing architectures are poor suited to the unique workload demands of deep learning, which has led to a surge in interest in memory-centric computing. Herein, a trilayer ($Hf_{0.8}Si_{0.2}O_2/Al_2O_3/Hf_{0.5}Si_{0.5}O_2$)-based self-rectifying resistive memory cell (SRMC) that exhibits (i) large selectivity (ca. $10^4$), (ii) two-bit operation, (iii) low read power (4 and 0.8 nW for low and high resistance states, respectively), (iv) read latency (<10 μs), (v) excellent non-volatility (data retention >$10^4$ s at 85 °C), and (vi) complementary metal-oxide-semiconductor compatibility (maximum supply voltage ≤5 V) is introduced, which outperforms previously reported SRMCs. These characteristics render the SRMC highly suitable for the main memory for memory-centric computing which can improve deep learning acceleration. Furthermore, the low programming power (ca. 18 nW), latency (100 μs), and endurance (>$10^6$) highlight the energy-efficiency and highly reliable random-access memory of our SRMC. The feasible operation of individual SRMCs in passive crossbar arrays of different sizes (30 × 30, 160 × 160, and 320 × 320) is attributed to the large asymmetry and nonlinearity in the current-voltage behavior of the proposed SRMC, verifying its potential for application in large-scale and high-density non-volatile memory for memory-centric computing.

[1] Division of Advanced Materials, Korea Research Institute of Chemical Technology (KRICT) 141 Gajeong-Ro, Yuseong-Gu, Daejeon, Republic of Korea. [2] Division of Materials Science and Engineering, Hanyang University, Seoul, Republic of Korea. [3] Intelligent Electronic Device Lab, Sahmyook University, Seoul, Republic of Korea. [4]These authors contributed equally: Kanghyeok Jeon, Jeeson Kim. ✉email: dooseokj@hanyang.ac.kr; kimgh@krict.re.kr

Recent trends in computation highlight a shift from conventional computing towards memory-centric computing. In conventional computing the processors are central, and the data subject to processing are transferred to the processors from a separate memory unit. Memory-centric computing avoids this data transfer through the memory hierarchy by placing processing power near the memory domain[1,2]. Examples of memory-centric computing include near-data-processing and in-memory processing (also known as processing-in-memory or computing-in-memory). A significant motivator for this shift is deep learning which requires immense memory capacity but simple data processing. Conventional computing exhibits significant shortcomings for this particular workload due to the enormous amount of data transferred between the separated memory and processors. These shortcomings include the memory wall, arising from the difference in performance between the processor and memory (processor > memory) and the consequent bottleneck in performance caused by the memory latency, and the immense power consumption over the buses between the processor and memory[3]. Specifically, the majority of the workload for deep learning results from the elementary operation for vector-matrix multiplication, which is a multiply-accumulate operation (one multiplication and one accumulation operation). Despite the simplicity of each operation, repetition over a massive matrix creates an enormous workload for the hardware because of its complexity $O(n^2)$. Notably, the trend in deep learning computing in recent years indicates an exponential increase in operation number; AlphaGo Zero in 2018 needed approximately 300,000 times the number of operations that were required for AlexNet in 2012. This trend is expected to continue. Therefore, employing memory-centric computing as a complementary approach or, more radically, an alternative approach to conventional computing is unavoidable if we wish to maintain sustainable progress in deep learning technologies.

Inference in deep learning only needs to read the weights in the memory, unlike training that needs to read and write the weights. Most of the workload for the hardware arises from inference rather than training as fully trained neural networks are only minimally re-trained and repeatedly applied to the given input data. Therefore, memory-centric computing for deep learning acceleration requires appropriate memories that have (i) large memory capacity, (ii) low latency in-memory read-out, (iii) low power consumption on memory read-out, (iv) non-volatility, and (v) complementary metal-oxide-semiconductor (CMOS) compatibility. Fast writing at low power is also desirable as a second priority. A common measure of hardware performance for inference is tera-operations per second per watt; therefore, requirements (ii) and (iii) directly improve the hardware performance. Requirement (i) is necessary because the state-of-the-art deep neural networks (DNNs) that can recognize real-world data are substantial in depth and unit number per layer. For instance, convolutional neural network (CNN)-based DNNs, such as, AlexNet[4], VGGNet (specifically, VGG-19)[5], GoogLeNet[6], ResNet (specifically, ResNet-152)[7], include approximately 60 M, 138 M, 4 M, and 60 M weights, respectively. When using a full precision float 32-bit format, the memory for the weights in a single model reaches 1.9 Gb, 4.4 Gb, 128 Mb, and 1.9 Gb, respectively. The memory should be sufficiently large to host these weights on-site to accelerate significantly the inference task. Requirement (iv) avoids loading the memory with massive amounts of weight data whenever it is rebooted. Compatibility with standard CMOS technologies (requirement (v)) is a critical criterion because the memory should be cointegrated with CMOS-based processing units.

Considering these requirements, there are several non-volatile memories which are regarded as potential storage-class memories

combining the advantages of main memories (random-access memory (RAM)) and data storage. They include ferroelectric RAM (FRAM)[8], spin-torque-transfer RAM (STT-RAM)[9], phase-change RAM (PcRAM)[10], and resistive RAM (RRAM)[11]. Thus far, these have not been commercialized as standalone storage-class memories due to a few shortcomings. For instance, FRAM and STT-RAM have an unavoidable limit to their memory capacity due to the use of transistors as bit-cell selectors and difficulty in fabrication. PcRAM may achieve high memory capacity using passive bit-cell selectors such as diodes and ovonic threshold switches; however, its high programming power and difficulty in multilevel programming preclude it from commercialization as storage-class memory. Similar to PcRAM, RRAM achieves high memory capacity and offers multilevel programming; however, its programming endurance is much lower than dynamic RAM and static RAM. Nevertheless, among these non-volatile memories, RRAM is the most likely candidate for the memory for memory-centric architecture for deep learning acceleration regarding requirements (i)–(iv) as the top priority. As a second priority, the advantages presented by RRAM in satisfying these requirements far outweigh its drawback of limited programming endurance. As a result, in-memory processors based on non-volatile memory frequently employ RRAM loaded with weight matrices[12–16].

RRAM offers feasible solutions to high memory capacity (requirement (i)) due to its multilevel operation and scalability down to the $4F^2$ design rule. Each memory bit-cell in RRAM is capable of multibit ($n$-bit; $n > 1$) representation using its $2^n$ resistance levels[17,18]. This significantly increases its memory capacity. Moreover, RRAM can be realized in passive crossbar arrays (CAs) where each bit-cell is formed at an intersection of a row- and column-line, meeting the ultimate $4F^2$ design rule[19–21]. Furthermore, the sneak current through unchosen cells, that leads to read-out errors should be considered[22,23]. Staking a passive selector on the memory cell at a cross-point avoids read-out errors only if it is possible for the selector to address just the chosen bits with negligible interference, similar to transistors in active CAs. However, because a single terminal is used to turn on the selector, read, and program the memory cell, the independent operation of each of the two series elements may be challenging unless the selector is specifically tailored to the memory cell or vice versa. An alternative is to use self-rectifying memory cells (SRMCs) which are single memory cells, whose highly nonlinear and asymmetric current–voltage ($I$–$V$) behavior alone enables the current sensing amplifier to distinguish between chosen and unchosen cells[24–30]. SRMCs have attracted significant attention because of their simplicity in bit-cell structure and thus potential compatibility with three-dimensional memory structure, enriching candidates for SRMCs, for example, $NbO_x/TiO_x/NbO_x$[26], $TiO_2/HfO_2$[29], $Ta_2O_5/HfO_{2-x}$[24], and Al-doped $HfO_2$[27]. Albeit excellent in most aspects, each has shortcomings that hinder it from being a promising candidate for an SRMC.

In this paper, we propose an $Hf_{0.8}Si_{0.2}O_2/Al_2O_3/Hf_{0.5}Si_{0.5}O_2$ trilayer-based SRMC that accurately meets the requirements for the main memory in memory-centric computing. Our proposed device has high selectivity (ca. $10^4$) and reliable 2-bit representation, which were verified in single cells in support of requirement (i), along with extremely low power consumption on a single read-out operation with 4 and 0.8 nW for low resistance state (LRS) and high resistance state (HRS), respectively, and latency of <10 μs in a single read-out operation in support of requirements (ii) and (iii). Moreover we also demonstrate the excellent non-volatility (data retention >$10^4$ s at 85 °C) in support of requirement (iv), and a programming pulse amplitude below 5 V, which is compatible with the CMOS voltage driving circuits in support of requirement (v). We summarize these features and

**Table 1 Performance comparison between our SRMC and previous results.**

| | Memory capacity | | Read power (nW) | | Read latency | Non-volatility | CMOS compatibility |
|---|---|---|---|---|---|---|---|
| | Selectivity ($I$ @$V_{op}$ /$I$ @-1/3$V_{op}$) | Multibit operation | LRS | HRS | | Retention | Max. supply voltage |
| **First priority** | | | | | | | |
| This work | ~$10^4$ | 2 bits | 4 | 0.8 | <10 µs | >$10^4$ s (cumulative) | ≤5 V |
| Haili et al. [24] | ~$10^4$ | – | 0.5 | $5 \times 10^{-3}$ | – | >$10^4$ s | – |
| Yoon et al.[25] | ~$10^4$ | – | 80 | $8 \times 10^{-3}$ | – | >$10^4$ | – |
| Kim et al.[26] | ~$2 \times 10^4$ | – | 0.6 | $3 \times 10^{-3}$ | – | >$10^4$ | – |
| Huang et al.[27] | ~$3 \times 10^2$ | – | 1.4 | 0.2 | – | >$10^4$ | ≤4 V |
| Zhou et al.[28] | ~$10^2$ | 2 bits | 0.3 | $2 \times 10^{-3}$ | – | >$10^4$ | ≤4 V |
| Hsu et al.[29] | ~$10^3$ | 2 bits | 2 | $2 \times 10^3$ | – | >$10^4$ | – |
| Chou et al.[30] | ~$3 \times 10^4$ | – | 1.2 | $2 \times 10^3$ | – | >$10^4$ | – |

| | Program power (nW) | Program latency | Program endurance | Test scale | Device structure |
|---|---|---|---|---|---|
| **Second priority** | | | | | |
| This work | 18 | 100 µs | >$10^6$ | 30 × 30, 160 × 160, 320 × 320 arrays | Ru/Hf$_{0.8}$Si$_{0.2}$O$_2$ /Al$_2$O$_3$/ Hf$_{0.5}$Si$_{0.5}$O$_2$/TiN |
| Haili et al.[24] | $6 \times 10^3$ | – | – | – | Pt/Ta$_2$O$_5$ /`HfO$_{2-x}$ /Hf |
| Yoon et al.[25] | 100 | – | ~$10^3$ (DC) | – | Pt/Ta$_2$O$_5$ /HfO$_{2-x}$ /TiN |
| Kim et al.[26] | 100 | – | ~$5 \times 10^3$ (DC) | – | Pt/NbO$_x$ /TiO$_y$/NbO$_x$ /TiN |
| Huang et al.[27] | $1.3 \times 10^3$ | 1 µs | ~$10^5$ | – | TiN /Al-HfO$_x$ /SiO$_2$/Si |
| Zhou et al.[28] | 4 | 10 ms | ~$5 \times 10^2$ | – | Cu/Al$_2$O$_3$ /aSi/Ta |
| Hsu et al.[29] | $8 \times 10^6$ | – | – | – | Ni/TiO$_2$ /HfO$_2$/Ni |
| Chou et al.[30] | $6 \times 10^3$ | – | 20 | 36 bit array | Ta/TaO$_x$ /TiO$_2$/Ti |

compare with findings from previous studies in Table 1. The selectivity value of each reference was extracted from the current ratio between at the maximum programming voltage and its negative one-third voltage.

## Results

**Resistive switching operation of unit SRMCs.** The proposed SRMC is based on an Hf$_{0.8}$Si$_{0.2}$O$_2$/Al$_2$O$_3$/Hf$_{0.5}$Si$_{0.5}$O$_2$ trilayer between a Ru top electrode (TE) and TiN bottom electrode (BE). The device fabrication procedure is detailed in the Methods Section. Figure 1a shows 90 $I$–$V$ hysteresis loops at 85 °C for 30 different SRMCs of 2 × 2 µm$^2$ area (three loops each). We chose a memory operation temperature of 85 °C which is the upper bound of the industrial temperature range (−40–85 °C). Notably, no electroforming was needed to activate the switching behavior. From the results, negligible cell-to-cell variability in $I$–$V$ behavior even at the elevated temperature was first identified. With the measured bipolar switching (BS) characteristics, both set (from HRS to LRS) and reset (from LRS to HRS) switching events are gradual under positive and negative voltage, respectively. The $I$–$V$ loops in Fig. 1a highlight large asymmetry in $I$–$V$ behavior between positive and negative voltage and large nonlinearity in $I$–$V$ on both sides and are eligible to be used as SRMCs. A voltage range that allows extremely low current (barely sufficient to distinguish between different resistance states) is referred to as an inhibit region; the large inhibit region (−0.8–0.6 V) of our SRMC is favorable to inhibit the sneak current through unselected cells in a passive CA. Additionally, given the gradual set switching

behavior, which is a self-compliance characteristic, no external current compliance is needed to protect the cell from a hard breakdown. This self-compliance characteristic is particularly desirable for passive CAs because a lack of transistors would otherwise limit current flow through memory cells.

We subsequently examined the BS of our SRMC in response to programming voltage pulses of different amplitudes (0–4.3 V) and widths (50 µs, 100 µs, and 1 ms). The measurement results in Fig. 1b indicate the onset of set switching at a positive voltage and a gradual reset behavior with the increase in reset voltage amplitude. The onset implies a threshold voltage for set switching, which enables non-destructive reading with a read-out voltage below this threshold voltage. Accordingly, we chose a read-out voltage of 2 V. The voltage pulses of 50 µs duration were insufficient to set the SRMC to a fully LRS, unlike the 100 µs and 1 ms duration cases (Fig. 1b), so we set the standard programming pulse width to 100 µs thereafter.

The high resistance, even in the LRS, causes a long RC time constant. This delimits the read-out speed significantly. We examined the read-out speed of the SRMC by applying a read-out pulse (2 V in amplitude and 5 µs in width) to five LRS SRMCs in parallel (Fig. 1c). It should be noted that we used five parallel SRMCs because the current level from a single SRMC is so small that it is barely measurable by an oscilloscope. The current plateau was reached after ~3 µs and the same delay was shown during discharging. Therefore, the read-out latency is below 10 µs.

The key to non-volatility is data retention at real memory-operating temperatures. Hence, we tested the stability of the LRS

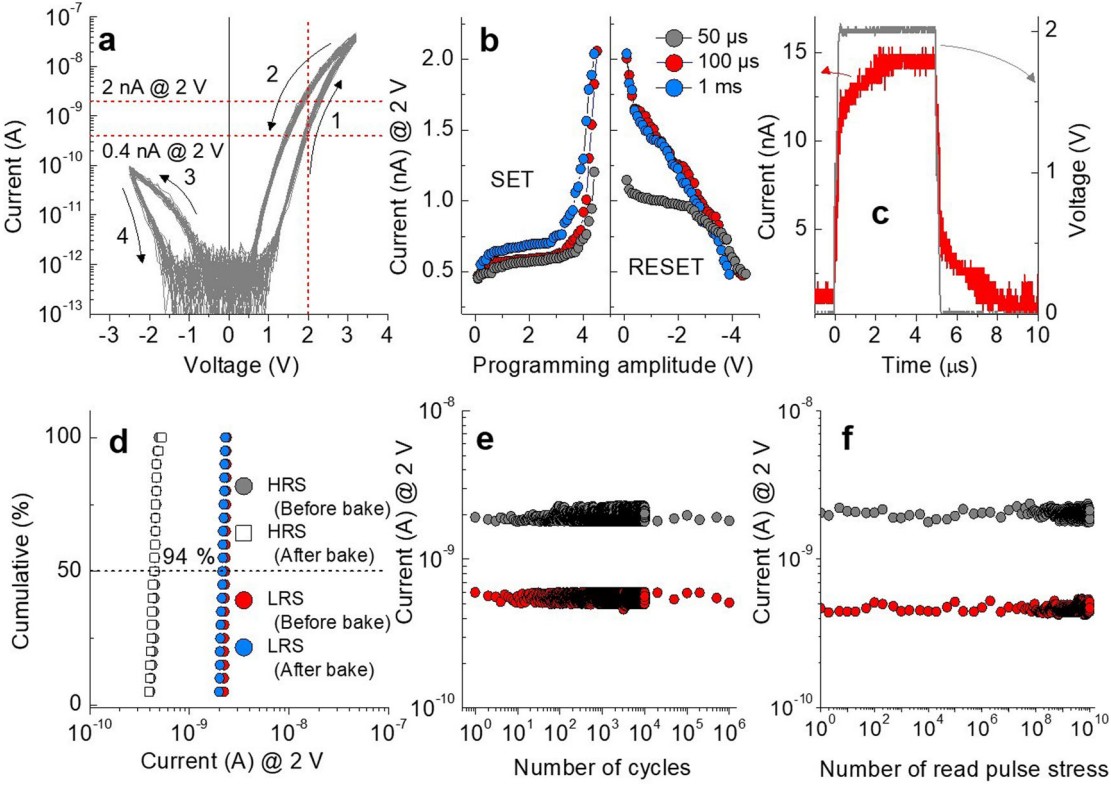

**Fig. 1 Electrical characterization of the unit self-rectifying resistive memory cell (SRMC). a** DC $I–V$ characteristics of 30 SRMCs. Arrows indicate switching direction. Readable current margin verified at 2 V is 0.4–2 nA. **b** Resistance states programmed by varying amplitude of programming voltage pulse for three pulse widths (50 μs, 100 μs, and 1 ms). **c** Read-out current in response to read-out pulse (2 V and 5 μs in amplitude and width, respectively). Current evaluated from voltage across 1 MΩ internal resistor of oscilloscope. **d** Memory retention of characteristic of 20 SRMCs in HRS and LRS as programmed and after baking (at 85 °C for 2 h). **e** Programming endurance of SRMC using 4.2 V/100 μs set and −4.3 V/100 μs reset pulses. **f** Read disturb characteristic of SRMC using repetitive reading pulse of 2 V/10 μs. (gray and red circle for LRS and HRS, respectively).

for 20 SRMCs maintained at the elevated temperature of 85 °C for 2 h. The 20 SRMCs were first programmed to the HRS using identical reset voltage pulses and their currents were read at 2 V. They were subsequently programmed to the LRS using identical set voltage pulses and the currents were read at 2 V. The 20 SRMCs in the LRS were heated up to 85 °C for 2 h, followed by current read-out at 2 V. The results in Fig. 1d indicate the excellent data retention even at the elevated temperature and almost negligible cell-to-cell variability in BS operation. Additionally, retention measurement at a higher temperature (125 °C for 2 h) on a single cell was also performed to confirm the stable data non-volatility as shown in Supplementary Fig. 1. We also identified the programming endurance of the SRMC for up to $10^6$ cycles, each with +4.2 V set and −4.3 V reset pulses (Fig. 1e). As elaborated in the Introduction section, because the number of *read* operation is much larger than that of writing operation in in-memory computing application, we examined read disturb characteristics of LRS and HRS by applying repetitive read pulses of 2 V. (10 μs width) (Fig. 1f) Up to $10^{10}$ of reading operation, our SRMC showed stable non-volatility in each resistance states, which largely exceeds the $10^6$ of endurance characteristic. (Fig. 1e)

**Structural analyses of SRMC.** Our SRMC is a vertical stack of Ru/ $Hf_{0.8}Si_{0.2}O_2$/$Al_2O_3$/$Hf_{0.5}Si_{0.5}O_2$ /TiN as confirmed by a cross-sectional high-resolution transmission electron microscope (HR-TEM) image (Fig. 2a). The upper $Hf_{0.8}Si_{0.2}O_2$ and lower $Hf_{0.5}Si_{0.5}O_2$ differ in chemical composition and are referred to as $HSO^1$ and $HSO^2$, respectively. $HSO^1$ and $HSO^2$ are separated by

a 1-nm-thick $Al_2O_3$ layer. These three layers are sandwiched between a Ru TE and TiN BE. Auger electron spectroscopy (AES) analyses on the SRMC consistently identify the stack structure as shown in Fig. 2b. Further analysis of the AES data indicates that $HSO^1$ and $HSO^2$ differ in chemical composition such that the cation-to-anion ratio is larger in $HSO^1$ than in $HSO^2$ (Fig. 2c). Additionally, we performed X-ray photoelectron spectroscopy analysis on our SRMC stack to compare the $HSO^1$ and $HSO^2$ layers (Fig. 2d–f). The two layers largely differ in the peak energy of an $O1s$ spectrum; the spectrum for $HSO^1$ peaks at approximately 530.4 eV while that for $HSO^2$ peaks at ~530.0 eV. The higher peak energy in $HSO^1$ indicates a higher concentration of non-lattice oxygen than in $HSO^2$[2,31,32]. Rutherford Backscattering Spectroscopy (RBS) measurement results shown in Supplementary Fig. 2 indicate that the chemical composition of $HSO^1$ and $HSO^2$ is $Hf_{0.8}Si_{0.2}O_2$ and $Hf_{0.5}Si_{0.5}O_2$, respectively.

**Resistive switching mechanism and current behavior of SRMC.** Regarding current transport in our SRMC, the current in both the LRS and HRS scales with a device area in the wide range 0.0484–100 μm² is plotted in Supplementary Fig. 3. This indicates interface-type switching as opposed to localized switching[33]; the whole device area is responsible for the switching by modulating the interfacial electronic energy barrier in a non-volatile manner[34,35]. This is consistent with the fact that our SRMC did not require an electroforming process, which is known to introduce conducting filaments[11,36]. In this regard, the largely asymmetric $I–V$ behavior may be due to the use of asymmetric metal electrodes and thus asymmetric interfacial barrier heights.

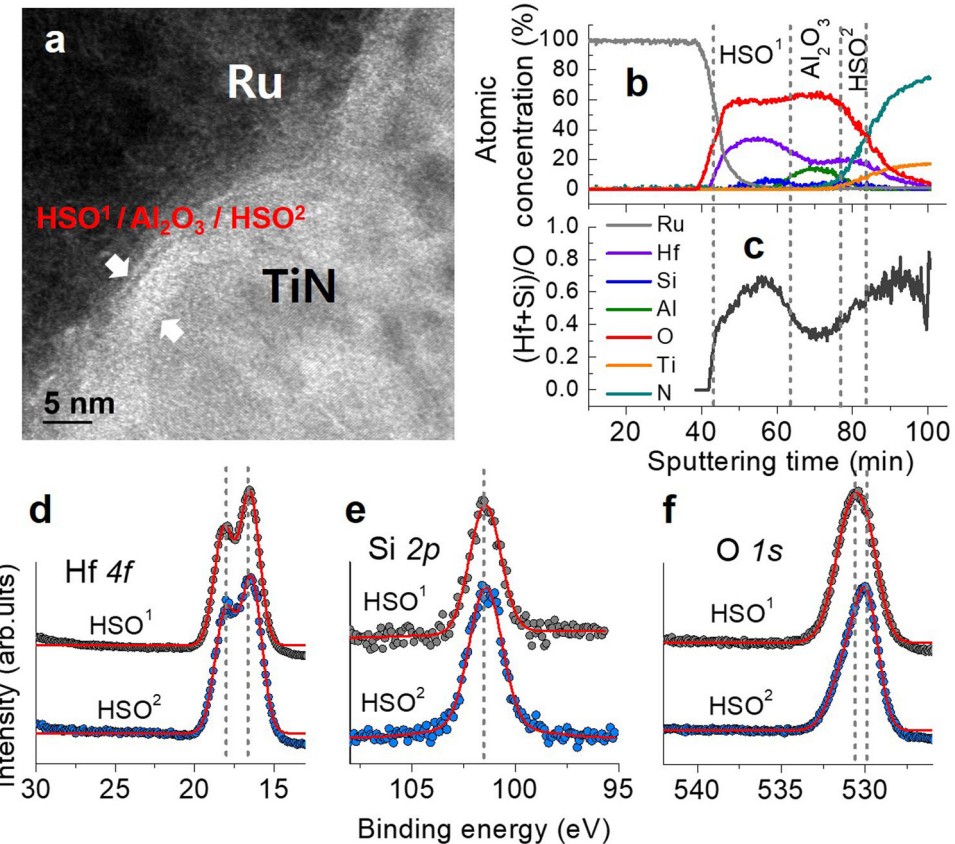

**Fig. 2 Microstructural and chemical analyses. a** Cross-sectional high-resolution transmission electron microscope image of our SRMC. **b** Depth profile of the elements, which were measured by Auger electron microscopy. **c** Atomic ratio (Hf+Si)/O along depth of SRMC. X-ray photoelectron spectra of **d** Hf4*f*, **e** Si2*p*, and **f** O1*s* emission for HSO[1] and HSO[2].

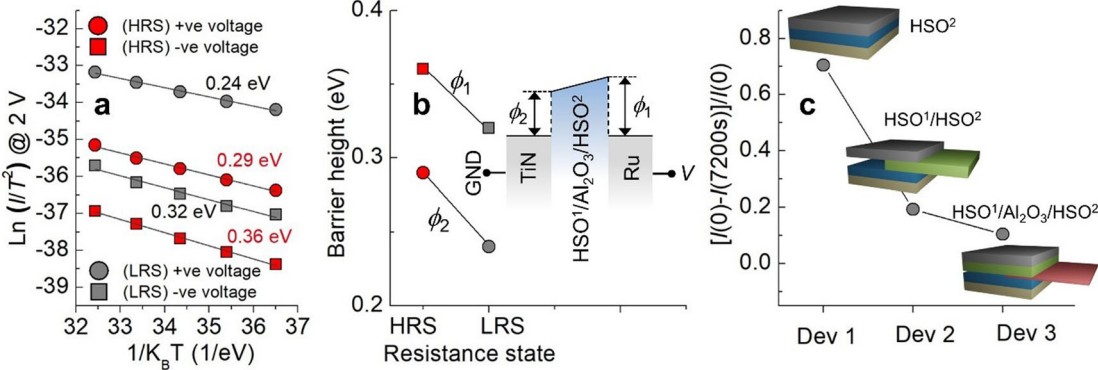

**Fig. 3 Current behavior in temperature domain emission equation and data retention for SRMC devices. a** Fitting Schottky emission equation to current measured at various temperatures (45–85 °C) and ±2 V for HRS and LRS. **b** Estimated barrier heights ($\phi_1$ and $\phi_2$) indicated in inset for HRS and LRS. **c** Data retention for the proposed SRMC (Dev 3) at 85 °C compared with Dev 1 (Ru/HSO[2]/TiN) and Dev 2 (Ru/HSO[1]/HSO[2]/TiN). The as-programmed current level and current level at 7200 s are denoted by $I(0)$ and $I(7200\ s)$, respectively.

The asymmetry in the interfacial barrier height was acquired by fitting the Schottky emission equation[37,38] to the experimental I–V data at different temperatures (45–85 °C). Here, the assumption was that the interfacial barrier at the cathode dictates the overall current transport through the SRMC such that the barrier limits the injection current level. The measured data on the ln($I/T^2$) and reciprocal $k_B T$ plane indicate good linearity, where $T$ and $k_B$ are absolute temperature and Boltzmann's constant, respectively (Fig. 3a). This analysis yields a barrier height pair, at the top and bottom interfaces, for the HRS and LRS. For both states, electron injection from the TE (i.e., under negative voltage), encounters a

higher Schottky barrier than the injection from the BE (i.e., under positive voltage), implying asymmetry in the barrier height due to asymmetry in the electrode presently in use (Fig. 3b).

The change in barrier height upon switching may be attributed to oxygen vacancy redistribution by the applied programming voltage[39,40]. Oxygen vacancies are redistributed in response to the direction of a programming field by electronic drift, resulting in the polarization of space charge. However, one should consider the high depolarization field built up during the programming period, which takes effect immediately after the removal of the programming field[40]. This presents a significant challenge for

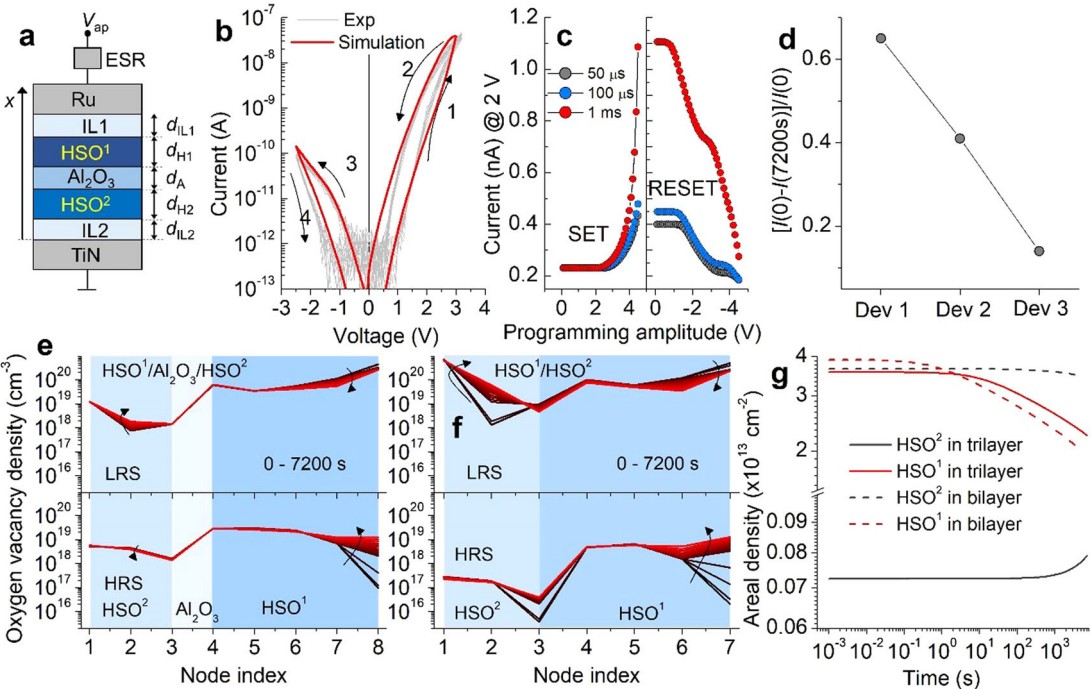

**Fig. 4 Resistive switching simulation. a** One-dimensional configuration of the SRMC for simulation. **b** Simulated I–V loop (quasi-static behavior) in comparison with experimental data. **c** Simulated switching behaviors in response to voltage pulses of different widths and amplitudes. **d** Simulated LRS retention for the HSO$^2$-only cell (Dev 1), HSO$^1$/HSO$^2$ cell (Dev 2), and HSO$^1$/Al$_2$O$_3$/HSO$^2$ SRMC (Dev 3). **e, f** Simulated oxygen vacancy distributions in the trilayer SRMC and HSO$^1$/HSO$^2$ cell in the LRS (upper panel) and HRS (lower panel). The change of the distribution in each state was monitored in the time range (0–7200 s). **g** Retention of areal density of oxygen vacancies in the LRS in each layer of the trilayer and bilayer cells.

data retention and thus non-volatility. The outstanding data retention in our SRMC may be achieved by the use of a separate oxygen reservoir (HSO$^1$) by an oxygen-blocking layer (Al$_2$O$_3$). Figure 2c shows that it is conceivable that the HSO$^1$ layer may have a higher oxygen vacancy concentration than the HSO$^2$ layer, serving as an oxygen reservoir, which creates excellent data retention. Data that support this hypothesis are presented in Fig. 3c; unlike the Ru/HSO$^1$/Al$_2$O$_3$/HSO$^2$/TiN stack, single-layer-based Ru/HSO$^2$/TiN stack creates a severe retention issue, identifying a current decrease by 70% at 85 °C. However, the key to high data retention in our SRMC lies not only in the oxygen reservoir (HSO$^1$) but also in the 1-nm-thick Al$_2$O$_3$ layer between HSO$^1$ and HSO$^2$. A comparison between the Ru/HSO$^1$/Al$_2$O$_3$/HSO$^2$/TiN and Ru/HSO$^1$/HSO$^2$/TiN SRMCs shows further improvement in data retention by inserting the Al$_2$O$_3$ layer between the HSO$^1$ and HSO$^2$ layers (Fig. 3c). Therefore, it is believed that the thin Al$_2$O$_3$ layer hinders the depolarization of space charge (oxygen vacancy)[41,42].

**Modeling of switching in SRMCs.** To identify the role of each layer in our SRMC, we modeled our SRMC from scratch regarding oxygen vacancy dynamics in response to the applied voltage. A schematic of the one-dimensional SRMC configuration is shown in Fig. 4a. We considered the trilayer as a mixed ionic-electronic conductor with free electrons and oxygen vacancies as mobile electronic and ionic defects. The oxygen vacancy redistribution in time and space domains was fully addressed using the Fick's second law. We used the quasi-static approximation to consider electron distribution in the SRMC given the large difference in diffusion coefficient between oxygen vacancy and electron. The defining features of the model are as follows.

- The electron transport is thermally activated such that the interfacial electron transport conforms to the Schottky

emission and the bulk electron transport to the band conduction rather than localized conduction.
- HSO$^1$ is given a lower reference state chemical potential $\mu_i^0$ for oxygen vacancy than HSO$^2$ to allow HSO$^1$ to hold a larger oxygen vacancy density than HSO$^2$ to be consistent with the experimental data in Fig. 2.
- The Al$_2$O$_3$ layer is given a lower oxygen vacancy diffusion coefficient than HSO$^1$ and HSO$^2$ by one order of magnitude.
- The Ru TE works as an oxygen vacancy source.
- An interfacial layer (IL) is placed at each interface, which may work as the Helmholtz layer[39,40].
- The breakdown of the first-order approximation (FOA) of the drift-diffusion equation is considered, which is likely the case when the internal electric field exceeds a few 10s MV/cm as for our SRMC.
- The experimentally observed asymmetry in I–V is realized by using asymmetric electrodes with different work functions ($W_{Ru} > W_{TiN}$).

The calculation procedure is elaborated in the Methods section.

In our model, the dc electronic current in the steady-state is dictated by the electronic injection current at the cathode, which conforms to the Schottky emission. That is, the injection current is determined by the interfacial electric field that lowers the Schottky barrier height (SBH) by image force. The redistribution of oxygen vacancies by programming voltage alters the interfacial electric field at both interfaces because the Debye length for the oxygen vacancy density considered is larger than the thickness of our trilayer. The relation between the interfacial electric fields and oxygen vacancy distribution is best explained using Poisson's equation.

$$\frac{dE}{dx} = \frac{q\rho(x)}{\epsilon_r\epsilon_0} \approx \frac{qc_{V_O}(x)}{\epsilon_r\epsilon_0},$$

where the space charge density $\rho$ is approximated to the oxygen

vacancy density $c_{Vo}$ because the free electron density is much lower than the vacancy density due to the large bandgap in the trilayer. The dielectric constant and vacuum permittivity are denoted by $\epsilon_r$ and $\epsilon_0$, respectively. That is, the trilayer is fully depleted. Solving the differential equation for the electric field at the bottom interface $E(0)$ or the top interface $E(d)$ yields the following equations.

$$\begin{cases} E(0) = -\frac{V_{ap}}{d} - \frac{q}{\epsilon_r \epsilon_0 d} \int_0^d \int_0^x c_{Vo}(x')dx'dx \\ E(d) = -\frac{V_{ap}}{d} + \frac{q}{\epsilon_r \epsilon_0} \int_0^d \left( c_{Vo}(x') - \frac{1}{d}\int_0^x c_{Vo}(x')dx' \right)dx \end{cases}, \quad (1)$$

where $d$ denotes the total thickness of the trilayer. From these equations, it is obvious that the change in vacancy distribution alters both interfacial electric fields. The key to non-volatile switching is that (i) set and reset switching operations cause $\triangle c_{Vo,t=0} \left( = c_{Vo,t=0}^{LRS} - c_{Vo,t=0}^{HRS} \right)$ sufficiently large to change $E(0)$ and $E(d)$ and (ii) the programmed distribution should be retained, $\triangle c_{Vo,t=0} \approx \triangle c_{Vo,t=\infty}$.

We took into account the breakdown of the FOA of the drift-diffusion equation in that the oxygen vacancy migration velocity exponentially increases with the electrochemical potential gradient[43–45]. The breakdown of the FOA may be the substrate for the voltage-time dilemma[46].

As boundary conditions, the TiN/HSO2 bottom interface forms oxygen-blocking contact while the Ru/HSO1 top interface allows oxygen vacancies to move through the interface (non-blocking contact) with a constant vacancy density on the Ru side of the interface. The parameters used in our modeling are listed in Supplementary Table 1, including several key parameters, e.g., vacancy diffusion coefficients in HSO1, HSO2, and Al2O3[47,48].

The response of our model to quasi-static stare-case voltage sweep (0.25 V/s) is plotted in Fig. 4b. The simulated $I$–$V$ loop is well consistent with the experimental data, identifying good reproducibility of experimental data in a quasi-static voltage domain. Subsequently, we tested the response of our model to voltage pulses of different widths (50 μs, 100 μs, and 1 ms) and amplitude (0.1–4.5 V). The results are shown in Fig. 4c. Similar to the experimental results, the set switching behavior represents the onset of switching at ~3 V, so that setting read-out voltage to 2 V was allowed as for the experimental measurements. The reset switching behavior (particularly, with 1 ms reset pulses) indicates a gradual change in resistance, in agreement with the experimental data.

The excellent LRS retention for our SRMC was successfully reproduced using our model as shown in Fig. 4d. We also modeled the other cells, HSO2-only cell (Dev 1) and HSO1/HSO2 cell (Dev 2) to identify their LRS retention characteristics. Note that for the two cells we used the same parameters as the SRMC model except their thicknesses. The comparison in Fig. 4d highlights the excellent LRS retention of our SRMC model in support of the experimental data.

As such, the key to data retention is the time-dependent redistribution of oxygen vacancies in each state. Our model simulation allows us to monitor the evolution of oxygen vacancy distribution at any time step. We set and reset the model and kept track of vacancy distribution for 7200 s. The monitored results are plotted in Fig. 4e; the upper and lower panels show the distributions for the LRS and HRS, respectively. The distributions indicate (i) the lower vacancy density in HSO2 than HSO1 in both resistance states, (ii) small change in vacancy density in both states over time, i.e., small $c_{Vo,t=0} - c_{Vo,t=7200}$ in both layers, and (iii) small difference in vacancy density between LRS and HRS, i.e., small $\triangle c_{Vo,t=0}$ in HSO2. Considering the indication (i), the resistance state of our model is mainly dictated by the oxygen vacancy density in HSO1 rather than HSO2. This is because the

interfacial electric fields are mainly determined by large space charge density as shown in Eq. (1); integrating the oxygen vacancy density over HSO2 is much smaller than over HSO1. The indication (ii) is the direct cause of the excellent LRS retention.

The indication (iii) is caused by the Al2O3 oxygen-blocking layer. The low diffusion coefficient of Al2O3 hinders oxygen vacancies from entering into (leaving from) HSO2 during set (reset) switching. This can be seen in comparison with the HSO1/HSO2 cell whose oxygen vacancy distributions in both states are plotted in Fig. 4f. Figure 4f identifies that the lack of the oxygen-blocking layer allows a large number of oxygen vacancies to enter into (leave from) HSO2, unlike the trilayer. Thus, the role of the Al2O3 oxygen-blocking layer in switching is to confine the active switching region to HSO1. The better LRS retention for the trilayer than HSO1/HSO2 is understood in terms of the confined switching to HSO1. According to Eq. (1), the larger the oxygen vacancy density, the larger the interfacial electric field evolves; the larger electric field in the vicinity of the top interface $E(d)$ drives the more oxygen vacancies back to the source (Ru). The unconfined cell (HSO1/HSO2) holds the more oxygen vacancies over than the trilayer, and the larger $E(d)$ releases the more oxygen vacancies to the vacancy source. To show this, we evaluated the areal density of oxygen vacancies in each layer for the trilayer and HSO1/HSO2 bilayer. The areal density was calculated by integrating the vacancy density over the HSO1 or HSO2 region. The results are shown in Fig. 4g, which identifies the larger decay in oxygen vacancy density in HSO2 in the bilayer than the trilayer.

**Two-bit operation of SRMCs**. The capability of multilevel programming was identified for four resistance levels: one HRS and three LRSs (L1, L2, and L3). The available resistance state ranges from 5 to 1 GΩ at a read-out voltage of 2 V, corresponding to 0.4–2 nA (Fig. 1b). The range was equally divided into four resistance bits, each with one of the four resistance states (0.4–0.8 nA for HRS, 0.8–1.2 nA for L1, 1.2–1.6 nA for L2, and 1.6–2 nA for L3). Each range (0.4 nA in width) was sub-divided into the available current range (0.16 nA) and forbidden range (0.24 nA) to reduce the state overlap probability (SOP) between neighboring states. We applied the incremental step pulse programming (ISPP)/error check and correction (ECC) method[17,18] considering the available current range (0.16 nA in width) which is separated from neighboring states by the forbidden range (0.24 nA in width). The multiple resistance levels were programmed using two distinct schemes: (i) erase-and-program and (ii) erase-free schemes. The former fully erases the SRMC from the LRS before each reprogramming, whereas the latter programs a new LRS directly from its current state without the full erase process. The erase-free scheme reduces the time complexity in multilevel programming because the erase process is omitted. These methods are shown in detail in previous studies[17].

To address the reliability of multilevel programming, 5 SRMCs were programmed into the four distinct resistance states at 85 °C using each programming protocol. Figure 5a identifies the multilevel operation of the five SRMCs (indexed #1–#5) using the erase-and-program scheme. Each subplot in Fig. 5a shows switching between the HRS and one of the three LRSs (L1, L2, and L3) over 50 cycles for one of the five SRMCs (#1–#5). The erase-free scheme was also applied to another set of five SRMCs subject to switching between L1 and L2, L2 and L3, and L1 and L3 over 50 cycles at 85 °C (Fig. 5b). During ISPP/ECC, the pulse amplitude required to program one of the three LRSs varied. We plotted the cumulative distribution of the pulse amplitudes for L1, L2, and L3 in Fig. 5c. The four measured resistance states were statistically analyzed to identify the SOP between the states as

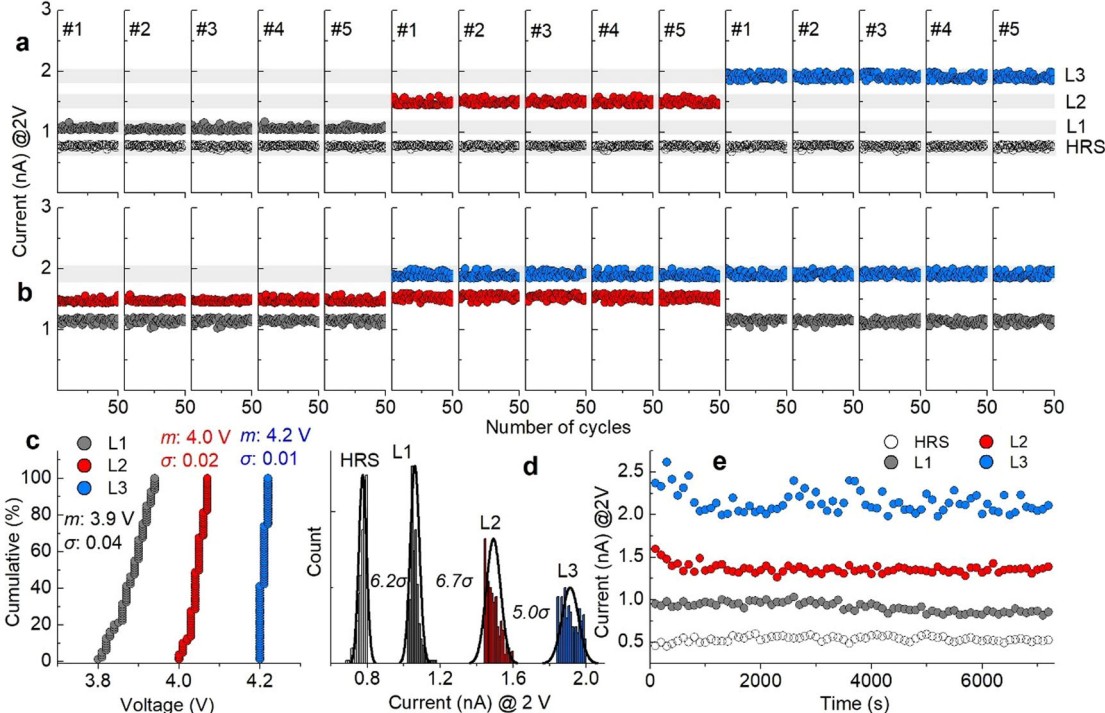

**Fig. 5 Two-bit states of SRMC.** Two-bit states programmed using **a** erase-and-program scheme and **b** erase-free scheme on five SRMCs (indexed #1–#5). **c** Cumulative distribution of amplitudes of two-bit programming pulses. Average amplitude and standard deviation denoted by $m$ and $\sigma$. **d** SOP between two-bit states. **e** Retention of two-bit states at 85 °C.

shown in Fig. 5d. The minimum distance in current between neighboring states arises from L2 and L3, which are separated by $5.0\sigma$ in a Gaussian distribution. This implies a $2.86 \times 10^{-5}\%$ probability of errors in a multibit read-out. Additionally, retention of the 2-bit data is an important concern. We addressed the retention by monitoring the four resistance states at 85 °C for 2 h, yielding the profiles of read currents in Fig. 5e. The data indicate a barely noticeable change in the current level for the four states even at the elevated temperature.

**SRMCs in a passive crossbar array**. We fabricated a $30 \times 30$ CA of the SRMCs, each of which was fully addressable. The layouts of the CA and morphology of a single SRMC are shown in Fig. 6a, b, respectively. To address a single cell, we applied an operation voltage ($V_{op}$) to the cell column-line (biasing line) with the row-line (ground line) being grounded. The current was measured on the ground line. Additionally, the other column- and row-lines were pulled up to the column- and row-inhibit voltage, respectively, to suppress the sneak current. We considered two biasing schemes: half-biasing (Scheme 1) and one-third biasing (Scheme 2). When addressing a cell, Scheme 1 pulls up the biasing line and half pulls up the column- and row-inhibit-biasing lines, whereas Scheme 2 pulls up the biasing line, pulls up the column-inhibit-biasing lines one-third, and pulls up the row-inhibit-biasing lines two-thirds. Schemes 1 and 2 are summarized in Table 2.

One hundred different SRMCs in the $30 \times 30$ CA were randomly chosen to characterize their I–V loops using Schemes 1 and 2, illustrated in Fig. 6c, d, respectively. The aim was twofold: the identification of disturbance from the unchosen cells and cell-to-cell variability of switching behavior. For each scheme, three I–V loops for each of 100 cells, i.e., 300 loops in aggregate, are shown in Fig. 6e, f. First, a comparison between Fig. 6e (6 f) and Fig. 1a identifies negligible effects of the 899 parallel SRMCs on the selected SRMC, leveraging their self-rectifying and highly

nonlinear I-V characteristics. Second, the appended I–V loops show negligible variability. The variability was evaluated by statistical analysis on the read-out currents (at 2 V) of the 100 cells (Fig. 6g, h for Schemes 1 and 2, respectively). The distributions for both schemes highlight good cell-to-cell uniformity in the $30 \times 30$ CA, and thus no overlap between HRS and LRS.

The excellent uniformity in the I–V loop is observed; this is due, in part, to the lack of electroforming, which is otherwise likely to endow each cell with uncontrollable random variability. Additionally, the current level in the inhibit region is comparable to the open circuit current level, implying extremely low current in the inhibit region, which is desirable when the CA size becomes large. In this CA configuration, each of the SRMCs is classified as (i) a selected cell between the biasing and ground lines, (ii) an unselected cell either between the column-inhibit-biasing and ground lines or between the biasing and row-inhibit-biasing lines, or (iii) an unselected cell between the column- and row-inhibit-biasing lines. The last two groups are named unselected groups 1 and 2, respectively. For each scheme, the theoretical voltage across an SRMC (voltage on a column-line minus voltage on a row-line) in each group is indicated in Fig. 6c, d. As shown in Fig. 6c, Scheme 1 allows half the pull-up voltage across the unselected group 1 cells (blue-filled circle) and zero voltage across the unselected group 2 cells (black-filled circle). Scheme 2 applies a positive one-third of the pull-up voltage (blue-filled circle) and negative one-third of the pull-up voltage (black-filled circle) to the unselected group 1 and 2 cells, respectively (Fig. 6d). Given a square CA ($N \times N$), the number of cells in the unselected groups 1 and 2 is proportional to $N$ and $N^2$, respectively. Thus, the larger the array, the more dominantly the unselected group 2 cells contribute to the resistance parallel to the selected cell's resistance. Although the effect of the unselected group 2 cells is barely noticeable in the $30 \times 30$ CA, it is likely to take effect in larger CAs, which is a challenge.

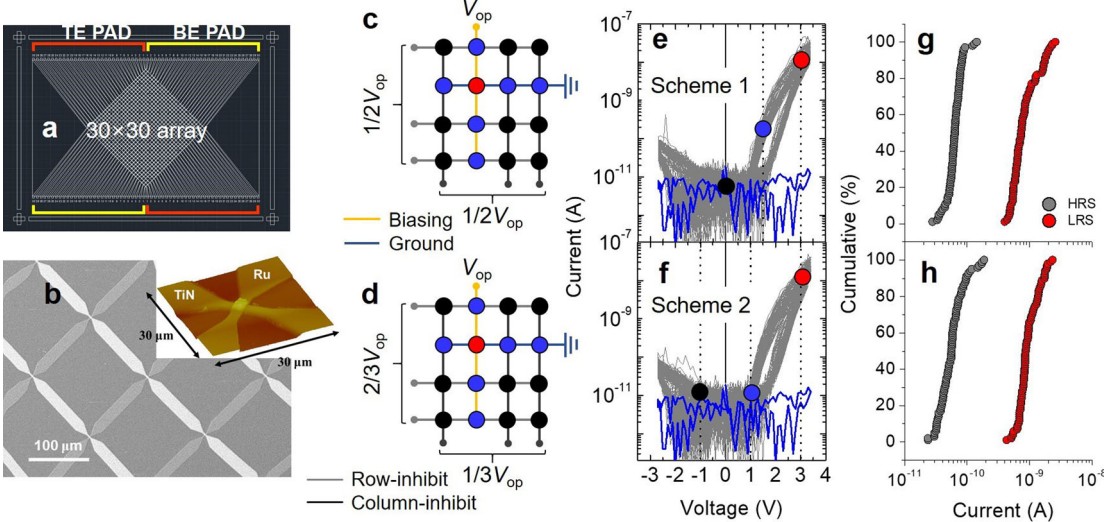

**Fig. 6 30 × 30 CA of SRMCs. a** Top view of CA layout. **b** Scanning electron microscope image of the array. The inset shows an atomic force microscope image of a unit SRMC. Schematic of **c** Scheme 1 and **d** Scheme 2. Appended *I–V* loops of 100 randomly chosen SRMCs (three loops for each SRMC), which were measured using **e** Scheme 1 and **f** Scheme 2. For $V_{op}$ = 3 V, voltage across selected cell (red-filled circle), unselected group 1 cell (blue-filled circle), and unselected group 2 cell (black-filled circle) is indicated. Open circuit current was also plotted (blue line). The currents read using Scheme 1 and Scheme 2 on the 100 SRMCs are shown in the distributions in **g** and **h**, respectively. Scheme 1: the mean current $m$ and standard deviation $\sigma$ for HRS and LRS are ($6.5 \times 10^{-11}$ A, $1.8 \times 10^{-11}$) and ($9.3 \times 10^{-10}$ A, $5.1 \times 10^{-10}$), respectively. Scheme 2: ($6.2 \times 10^{-11}$ A, $3.2 \times 10^{-11}$) and ($1.0 \times 10^{-9}$ A, $4.1 \times 10^{-10}$) for HRS and LRS, respectively.

**Table 2 Voltage across cells over a CA for Schemes 1–4.**

|  | Scheme 1 | Scheme 2 | Scheme 3 | Scheme 4 |
|---|---|---|---|---|
| Biasing line voltage | $V_{op}$ | $V_{op}$ | $V_{op}$ | $V_{op}$ |
| Row-inhibit-line voltage | $1/2V_{op}$ | $2/3V_{op}$ | $1/3V_{op}$ | $1/3V_{op}$ |
| Column-inhibit-line voltage | $1/2V_{op}$ | $1/3V_{op}$ | $2/3V_{op}$ | $1/3V_{op}$ |
| Voltage across a selected cell | $V_{op}$ | $V_{op}$ | $V_{op}$ | $V_{op}$ |
| Voltage across an unselected group 1 cell | $1/2V_{op}$ | $1/3V_{op}$ | $2/3V_{op}$ | $2/3V_{op}$ $1/3V_{op}$ |
| Voltage across an unselected group 2 cell | 0 | $-1/3V_{op}$ | $1/3V_{op}$ | 0 |

To address this challenge, we investigated the *I–V* behavior of a predefined selected SRMC embedded in a 160 × 160 (~25 kb) and 320 × 320 (~100 kb) CA. The layouts of the CAs are shown in Supplementary Fig. 4. Note that these arrays were not random-accessible because the number of lines exceeds the number of currently available probes. Instead, we programmed the unselected group 2 cells simultaneously to LRS by leaving all row-lines (except the signal row-line) and all column-lines (except the signal column-line) common. The measurement configuration is depicted in Supplementary Fig. 5. The detail of the measurement is written in the Methods section. Schemes 1 and 2 also applied to the large CAs. For both schemes, the voltage across the selected SRMC and unselected groups 1 and 2 cells is indicated on the *I–V* loop taken from Fig. 1a in Fig. 7a, b. Two additional biasing schemes (Schemes 3 and 4) were considered for comparison. Scheme 3 applies a one-third $V_{op}$ and two-thirds $V_{op}$ to the row- and column-inhibit lines, respectively. Therefore, the voltage across the unselected group 1 and 2 cells is two-thirds $V_{op}$ and one-third $V_{op}$, respectively (Fig. 7c). Scheme 4 applies a one-third $V_{op}$ to both the row- and column-inhibit lines, allowing one-third or two-thirds $V_{op}$ across the unselected group 1 cell and zero voltage across the unselected group 2 cell (Fig. 7d). The details of Schemes 3 and 4 are summarized in Table 2.

To examine the sneak current from the unselected groups 1 and 2 cells, we attempted to program all unselected groups 1 and 2 cells into their LRS and subsequently examined the *I–V* characteristics of the selected cell. The measured *I–V* loops for the selected cell embedded in the 160 × 160 CA are shown in Fig. 7e. The different biasing schemes caused a negligible difference in the *I–V* loops of the selected cell. This confirms that the self-rectifying and nonlinear *I–V* behavior of the SRMC maintains a sufficiently low current through the unselected cells to enable the true current to be read through the selected cell.

The 320 × 320 CA allows the sneak current to vary the ground line current more obviously than the smaller CAs, yielding more obviously distinct *I–V* loops depending on the voltage-application scheme (Fig. 7f). Scheme 2 yields a lower current than Scheme 1 over the whole voltage range, whereas the largest current level was yielded by Schemes 3 and 4. This is because Scheme 2 applies the lowest voltage to unselected group 1 cells, which share the same row-line as the selected cell. Nevertheless, the switching behavior of the selected cell indicates two distinct states despite the sneak current in this seemingly worst-case conductance distribution.

Two-bit operation of SRMCs in the CAs was examined successfully. Owing to the random-accessibility of the 30 × 30 CA, five SRMCs were chosen randomly and subject to the two-bit operation, resulting in readable four states (Supplementary Fig. 6) as for the single cells. Additionally, we identified the two-bit operation of the predefined SRMC in the 320 × 320 CA, yielding clearly distinct four states (Supplementary Fig. 7).

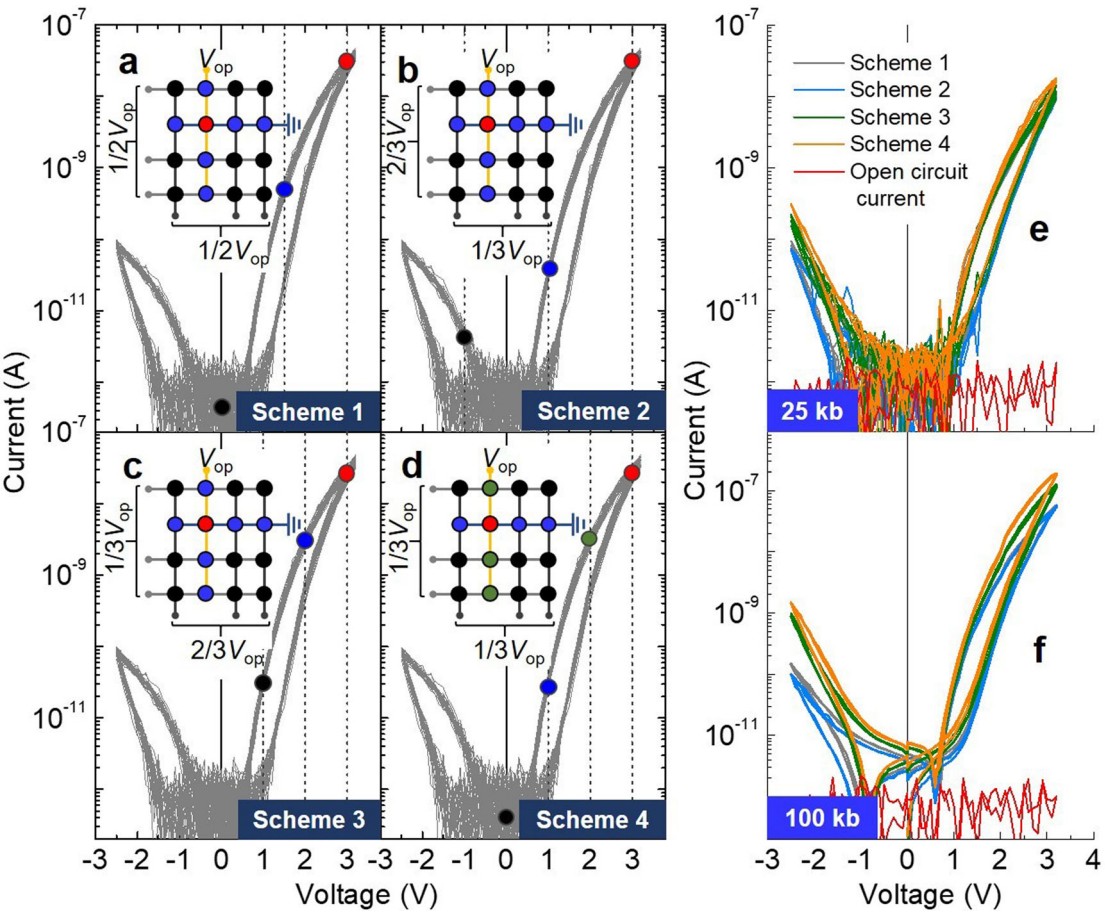

**Fig. 7 160 × 160 and 320 × 320 CAs of SRMCs. a–d** Illustrations of Schemes 1–4 and voltage across different cells indicated by different colors. *I–V* loops of selected cell that was embedded in **e** 160 × 160 and **f** 320 × 320 CA.

**Vector-matrix multiplication acceleration using the 30 × 30 crossbar array.** Finally, we identified the feasible acceleration of vector-matrix multiplication ($\mathbf{w} \times \mathbf{x}$; $\mathbf{w} \in \mathbb{Z}^{30 \times 30}$, $\mathbf{x} \in \mathbb{Z}^{30}$) by reducing the computational complexity to $O(n)$. To this end, we aimed to calculate a dot product $\mathbf{w}[i, :] \cdot \mathbf{x}$ at one cycle, where $\mathbf{w}[i, :]$ denotes the $i$th row of matrix $\mathbf{w}$. We restricted the elements $w$ of matrix $\mathbf{w}$ to 2-bit integers ($w \in \{0, 1, 2, 3\}$) and the elements $x$ to 1-bit integers ($x \in \{0, 1\}$). The matrix $\mathbf{w}$ was transposed and mapped onto our 30×30 SRMC CA (conductance of each cell $\in \{HRS, L1, L2, L3\}$). The vector $\mathbf{x}$ with 1-bit integer elements was encoded as a voltage array $\mathbf{V}_{ap}$ ($V_{ap} \in \{0, 2V\}$) and applied to the 30 row-lines of the CA (Fig. 8a). The current measured at the $i$th column-line was the intermediate result of the dot product $\mathbf{w}[i, :] \cdot \mathbf{x}$. As depicted in Fig. 8b, we addressed one column at one cycle by pulling down the chosen column-line to the ground while inhibit voltages ($V_{inhibit}$) were applied to the rest of column-lines ($2/3 V_{ap}$), so that we reduced the complexity to $O(n)$, which is otherwise $O(n^2)$.

We chose four random matrices ($w_1$, $w_2$, $w_3$, and $w_4$) of different sparsities (0, 25, 51, and 55%, respectively). The percentage of each integer (0, 1, 2, 3) in each matrix is shown in Fig. 8c. The chosen matrices were mapped onto four 30 × 30 CAs such that the individual cells of the CAs were randomly accessed and programmed to the correct conductance states using Scheme 2. The programmed conductance map for each matrix is shown in Fig. 8d–g. The conductance of each SRMC was individually read out at a read-out voltage of 2 V to acquire the maps. We then performed the dot product $\mathbf{w}[i, :] \cdot \mathbf{x}$ for each $i$ at

one cycle with vector $\mathbf{x}$ of ones, i.e., $\mathbf{x} = [1, 1, …, 1]$. The vector-matrix multiplication operation for each matrix thus consumes 30 column-line-addressing cycles, yielding a current vector $\mathbf{j}$ ($\in \mathbb{R}^{30}$) as the intermediate product (Fig. 8d–g). The measured current at each column-line is almost identical to the current value extrapolated from each cell current in the same column, indicating marginal disturbance from the unselected cells. For the multiplication with four matrices ($\mathbf{w}_1$, $\mathbf{w}_2$, $\mathbf{w}_3$, and $\mathbf{w}_4$), the CA domain consumes powers of 4.22, 3.44, 2.83, and 2.69 μW, respectively. The considered multiplication is the worst case in terms of power consumption because of the extremely dense vector $\boldsymbol{x}$ (of ones).

To output the final product $\mathbf{z}$ ($\mathbf{z} = \mathbf{w} \times \mathbf{x}$; $\mathbf{z} \in \mathbb{Z}^{30}$), current from the $i$th column $j_i$ for all $i$ needs to be encoded as a binary number, which subsequently enters into the near-memory digital domain for additional processing. A common method is to convert the summed current to voltage and subsequently to quantize the converted voltage using an analog-to-digital converter (ADC)[49]. Alternatively, the summed current can directly be converted to a binary value using a current sense amplifier (CSA) with multiple reference currents that are iteratively compared with the summed current[50]. In either way, the important consideration is twofold: (i) energy consumption and (ii) bit-width of the product $\mathbf{z}$. Regarding power consumption, ADCs are well known to consume a considerable amount of energy insomuch as the total energy consumption of an RRAM-based inference accelerator is dominated by the ADCs[13]. An alternative method using a CSA[50] keeps the static current from the chosen line flowing while the

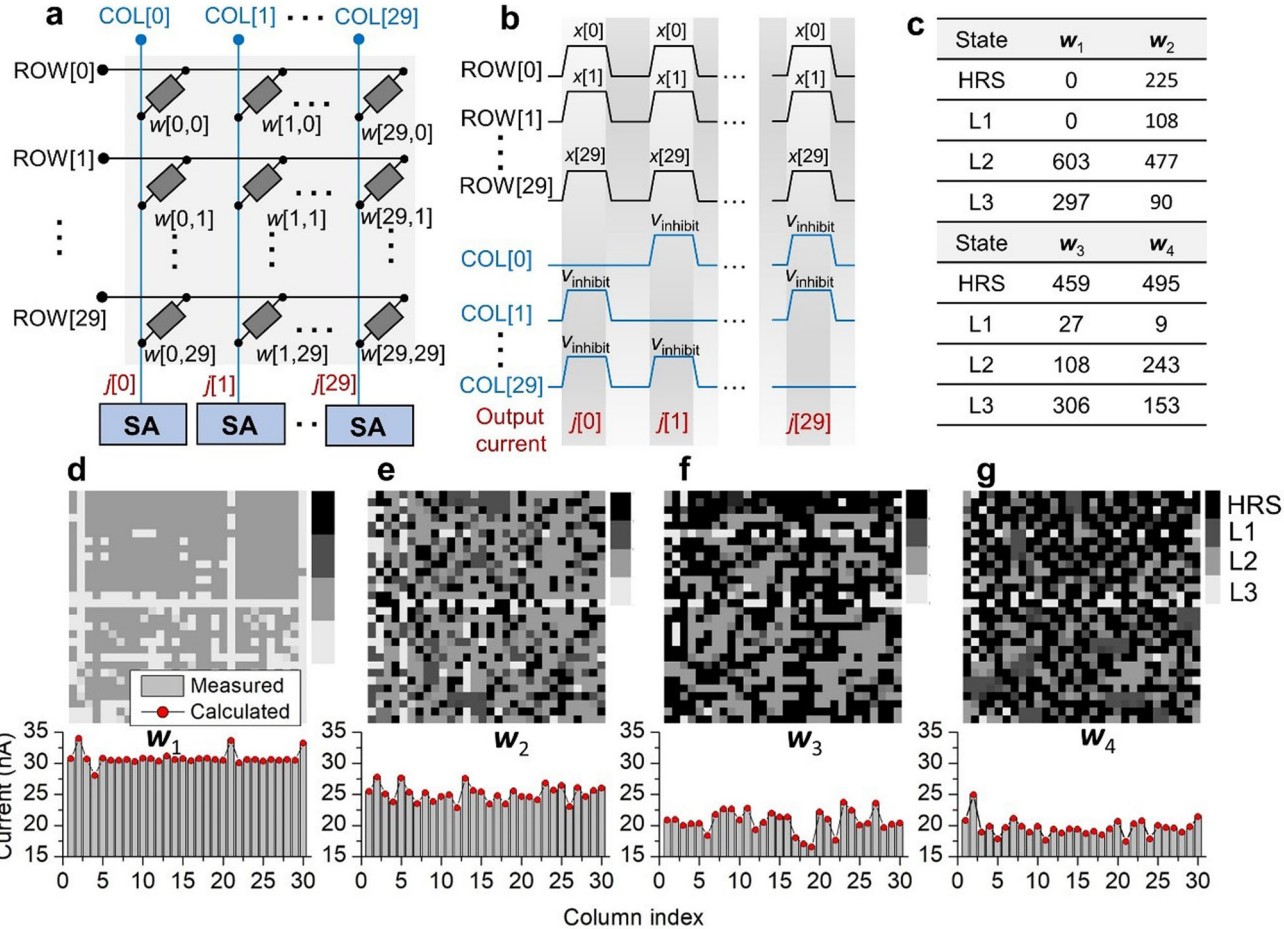

**Fig. 8 Acceleration of vector-matrix multiplication using the 30 × 30 CA. a** Configuration of a 30 × 30 matrix w mapped onto a CA of the same size. Vector x is encoded as voltage signals ('0' = 0 V, '1' = 2 V) and enters into the row-lines (ROW[0]–ROW[29]). The resulting current vector j as an intermediate product enters into sense amplifiers (SAs) to be quantized. **b** Schematic timing diagrams of row- and column-line signals. The inhibit voltages applied to unchosen column-lines are denoted by $V_{inhibit}$. **c** Statistics of four states (HRS, L1, L2, L3) in four random matrices ($w_1$–$w_4$). **d**–**g** (upper panel) Conductance maps of the four random matrices ($w_1$–$w_4$) and (lower panel) measured current vectors j for the four matrices. We considered a vector x of ones. The measurement results are compared with the calculated current vectors using the measured currents on individual cells.

summed current being converted iteratively, causing additional energy consumption. The bit-width should be chosen carefully to avoid the performance, i.e., inference, degradation by the quantization bit-width. As shown in quantized neural networks such as DoReFa-Net[51], the resolution, i.e., bit-width, of activations more critically determines the inference accuracy than that of weights. The activation resolution is dictated by the bit-width of the output z. Therefore, the bit-width of the product **z** is an important consideration in the design of summed current-encoding circuits.

Regarding multibit factor **x**, time-division multiplexing is a desirable method by encoding the vector **x** as shown in Fig. 9. Because of the nonlinear $I$–$V$ behavior in the LRS of our SRMCs, encoding a factor as input voltage amplitude is unsuitable unlike linear $I$–$V$ cases[14,52]. The $l$-bit elements $\mathbf{x}[i]$ are time-division multiplexed from the least significant bits (LSBs) to the most significant bits (MSBs) and are applied to the row-lines at one column-line addressing cycle for the dot product $\mathbf{w}[i, :] \cdot \mathbf{x}$. Thus, each dot product cycle includes $l$ sub-cycles. The output current at each sub-cycle is encoded as a binary value and subsequently multiplied by $2^{k-1}$, where $k$ denotes the digit corresponding to the sub-cycle. The results are finally summed to output the dot product $\mathbf{w}[i, :] \cdot \mathbf{x}$.

## Discussion

We proposed an SRMC based on a $Hf_{0.8}Si_{0.2}O_2/Al_2O_3/Hf_{0.5}Si_{0.5}O_2$ trilayer stack, which highlights large selectivity (~$10^4$), two-bit operation, low read power (4 nW for LRS and 0.8 nW for HRS), read latency (<10 μs), excellent data retention (>$10^4$ s at 85 °C), and CMOS compatibility (maximum supply voltage ≤5 V). Particularly, the large selectivity due to the high asymmetry and nonlinearity in the $I$–$V$ behavior potentially supports high-density passive CAs of the SRMCs, which is one of the key elements to memory-centric computing in support of deep learning acceleration. Feasibility was identified in 30 × 30, 160 × 160, and 320 × 320 arrays of our SRMCs. The $I$–$V$ behavior of an isolated SRMC was reproduced well in the arrays without significant effects on the unselected cells. These excellent characteristics may be attributed to nonfilamentary switching, i.e., switching on the grounds of Schottky barrier modulation at the cathode, which is homogeneous over the device area. A common issue of such nonfilamentary switching is data retention due to the rapid depolarization of point defects, which was overcome by using the engineered trilayer switching stack in this study. Furthermore, the low programming power (ca. 18 nW), latency (100 μs), and endurance (>$10^6$) highlight the energy-efficiency and highly reliable random-access memory of our SRMC.

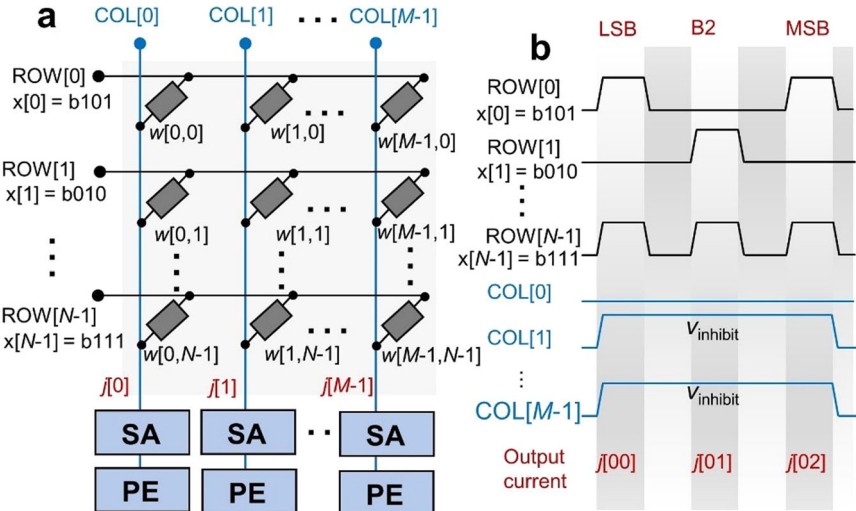

**Fig. 9 Acceleration of multibit vector-matrix multiplication. a** Configuration of a mapped weight matrix w ($M \times N$) and multibit vector x (here, 3-bit). Elements x[i] are time-multiplexed, so that the multiplication delay is proportional to the bit-width of elements x[i]. **b** Timing diagrams of the signals to calculate w[:,i]·x for a given i with multibit elements including x[0] (=b101), x[1] (=b010), and x[29] (=b111). The resulting currents in the three time divisions (j[00], j[01], and j[02]) are first quantized by the SAs and subsequently multiplied by 1, $2^1$, and $2^2$, respectively, and summed in the processing elements (PEs).

Ideally, CAs may achieve the ultimate complexity $O(1)$ of vector-matrix multiplication beyond the complexity $O(n)$ by addressing all column-lines at one cycle. The basic premise is that all non-ideal factors, e.g., sneak current and line resistance effects, are excluded. The sneak current effect may be marginal because all bit-cells are supposed to be non-negatively biased when all column-lines are simultaneously addressed, i.e., grounded. However, the effect of finite line resistance is significant. The finite line resistance causes the inhomogeneous distribution of bit-cell voltages over the cells on the same row-line such that the further a bit-cell from the row-line contact, the lower voltage is applied across the bit-cell. Further, this effect is boosted when the bit-cells on the same row-line allow simultaneous current flow, which is the case of all column-line addressing.

Additionally, simultaneously addressing all column-lines requires one CSA and following logic circuit per column line, whereas addressing one column-line at a cycle allows one CSA to be shared among a group of column-lines through time-division multiplexing. This additional peripheral circuit-area overhead can be prohibitive in large-scale CAs. Therefore, the complexity reduction to $O(1)$ may be realized only when these challenges are overcome.

## Methods

**Device fabrication.** A 200-nm-thick TiN layer was sputtered on a SiO₂/Si substrate and patterned to a shape of crossbar-type BE. The TiN BE was patterned by a conventional photolithography and dry-etching process by an inductively coupled plasma reactive ion etching (ICP-RIE). An ICP power of 200 W and a substrate bias power of 20 W were maintained during TiN etching. During the dry-etching process, the reactant gas flow rates of Ar and $Cl_2$ were maintained at 5 standard cubic centimeters per minute (sccm) and 30 sccm, respectively. Furthermore, the process temperature was maintained at 25 °C by a water-circulation cooling system. The observed etching rate was ~70 nm/min. The residual photo-resist (PR) on the patterned TiN BE was removed by an acetone etchant and cleaned sequentially with isopropyl alcohol and deionized water. The 1-nm-thickness $HSO^1$ and 2-nm-thickness $HSO^2$ thin films layer were then deposited by traveling wave-type ALD at 250 °C using a tetrakis-ethylmethylamido hafnium (TEMA-Hf) and bis(diethylamino)silane (BDEAS) precursor, respectively, and $H_2O$ and $O_2$ plasma as a source of Hf and Si oxidant, respectively. To form each of $HSO^1$ and $HSO^2$ layers, the super-cycle ALD processes with $HfO_2$ and $SiO_2$ layers were used. During the deposition of $HSO^1$ and $HSO^2$ thin film, the ALD cycle ratios of 1:3 and 1:1 for $SiO_2:HfO_2$ were set, respectively. Between the $HSO^1$ and $HSO^2$ layers, the $Al_2O_3$ thin film layer was deposited by traveling wave-type ALD at 150 °C using a trimethyl aluminum (TMA) and $H_2O$ as a source of Al and oxidant, respectively.

Subsequently, the crossbar-type TE pattern was formed by photolithography and then the 100-nm-thick Ru layer was deposited by DC magnetron sputtering. Finally, through the conventional lift-off process, the $Ru/HSO^1/Al_2O_3/HSO^2/TiN$ stacked device was fabricated. All of the unit and CA devices have identical fabrication processes.

**Structural analysis of SRMC device.** The sample for TEM analysis was prepared by a focused ion beam (FIB, Helios NanoLabTM by FEI) operation. HR-TEM (Tecnai G2 F30 S-TWIN by FEI) analysis was then performed to obtain a cross-sectional view of the $Ru/HSO^1/Al_2O_3/HSO^2/TiN$ stacked RS device. The XPS analysis was performed to examine the chemical binding status of the $HSO^1$ and $HSO^2$ layer with an X-ray photoelectron spectrometer (XPS, Thermo Fisher Scientific Inc.) using an Al $K\alpha$ source with a spot size of 400 μm and energy step size of 0.1 eV. The samples for XPS analysis were prepared using blanket-type HSO thin films on a TiN substrate. It should be noted that because the thicknesses of $HSO^1$ and $HSO^2$ in the device are very thin, additional samples were prepared for XPS analysis. The elemental depth profile was obtained using AES (ULVAC-PHI 700, coaxial full CMA type analyzer, 10 kV/10 nA of electron beam energy) measurements. During the AES measurement, a sputtering rate of ~0.2 Å/s was maintained. RBS (National Electrostatics Corp.) analysis for qualified chemical composition of our active layers was performed with separately prepared $HSO^1$ and $HSO^2$. (50-nm-thick each on SiO₂/Si substrate) AFM (Digital Instruments Dimension 3000, Veeco Science) analysis was conducted to observe the CA device morphology and determine the exact feature sizes of the RS device. The CAs were observed using a scanning electron microscope (SEM, JEOL JSM-6700F).

**Electrical measurements.** The resistive switching characteristic of the device was measured using an HP4145B semiconductor parameter analyzer in the $I-V$ sweep mode. Measuring temperature was controlled by a hot stage using a temperature controller. The pulse-based electrical measurements were conducted using an HP4145B, arbitrary function generator (Agilent 81150 A), oscilloscope (MSOX3024T, Tektronix), and electromechanical radiofrequency electric-circuit switch box. Throughout the measurement processes, the voltage was biased to the Ru TE, while the TiN BE was electrically grounded. The resistance (or current) values of the programming and erasing were verified at 2 V using the SPA. Measuring the 2-bit RS operation, ISPP/ECC algorithms were performed using two convertible electrical circuits composed of [AFG-RS device-OSC] and [SPA-RS device], respectively. These two types of electrical circuits were approached alternately by the electromechanical RF electrical circuit switch boxes. In the random-access operation (Fig. 6), the 30 × 30 sized matrix switching zig was additionally equipped in the previous electrical circuit. All electrical measurements were performed using a LabViEW™-based control program.

**Modeling of resistive switching dynamics.** We modeled the resistive switching behaviors of our SRMC regarding oxygen vacancy dynamics in response to the applied voltage. The one-dimensional model configuration considered is shown in Fig. 4a. The SRMC consists of five layers, $HSO^1/Al_2O_3/HSO^2$ plus two interfacial dipole layers between $HSO^1$ and TE ($IL_1$) and $HSO^2$ and BE ($IL_2$).

Note that we also considered an equivalent series resistor (ESR) with its resistance $R_{ESR}$. The voltage across the SRMC ($V_c$) at the applied voltage $V_{ap}$ is therefore expressed as

$$V_{ap} = -AR_{ESR}j + V_c, \qquad (2)$$

where $j$ and $A$ denote the electric current density through the whole circuit and SRMC area, respectively. According to the Kirchhoff's current law (KCL), the electric current density $j$ should be invariant along the spatial axis $x$. The current density $j$ consists of dc current density $j_{dc}$ and displacement current density as follows.

$$j = j_{dc} - \epsilon_r \epsilon_0 \frac{d}{dt}\left(\frac{dV}{dx}\right), \qquad (3)$$

where $\epsilon_r$ and $\epsilon_0$ denote the dielectric constant and permittivity of vacuum, respectively. The dc current density $j_{dc}$ consists of an electronic contribution $j_e$ and ionic contribution $j_{dd}^i$ because the SRMC is considered as a mixed ionic-electronic conductor (MIEC). The ionic (oxygen vacancy) current density $j_{dd}^{Vo}$ is attributed to the drift and diffusion of oxygen vacancies, which are driven by the gradients of internal electrostatic potential and chemical potential, respectively. The same holds for electronic dc current density $j_e$. Nevertheless, the much larger diffusion coefficient (and thus mobility) of electrons than that of oxygen vacancies allows us to use the quasi-static approximation in that the electronic distribution and current retain their equilibrium at given distributions of oxygen vacancies and internal potential across the SRMC at any given time.

Integrating Eq. (3) over $x$ at a given time $t$ yields

$$\frac{dV_c}{dt} = \int_0^{d_{tot}} (\epsilon_r \epsilon_0)^{-1} j_{dc} dx - j \int_0^{d_{tot}} (\epsilon_r \epsilon_0)^{-1} dx \qquad (4)$$

Note that the dielectric constant $\epsilon_r$ is a function of $x$ given the multilayer structure, and the total current $j$ is independent of $x$ according to the KCL. Differentiating Eq. (2) with time $t$ and subsequently entering Eq. (4) lead to

$$\frac{dV_{ap}}{dt} = -AR_{ESR}\frac{dj}{dt} + \int_0^{d_{tot}} (\epsilon_r \epsilon_0)^{-1} j_{dc} dx - j \int_0^{d_{tot}} (\epsilon_r \epsilon_0)^{-1} dx \qquad (5)$$

Equation (5) was numerically solved using the finite different method, which allows us to discretize Eq. (5) in the time interval $t_1 - (t_1 + \Delta t)$ as follows.

$$\Delta t^{-1} \left[V_{ap}(t_1 + \Delta t) - V_{ap}(t_1)\right] = -AR_{ESR}\Delta t^{-1}\left[j(t_1 + \Delta t) - j(t_1)\right] + B_1 - B_2 j(t_1 + \Delta t) \qquad (6)$$

$B_1$ in Eq. (6) is the integration of the dc current density $j_{dc}$ over axis $x$, which should be separately calculated across each layer because of the inhomogeneous dielectric constant distribution through the layers.

$$B_1 = \sum_i \sum_j (\epsilon_r^{(i)} \epsilon_0)^{-1} j_{dc}(t_1 + \Delta t) \Delta x^{(j)}, i \in \{IL_1, HSO^1, Al_2O_3, HSO^2, IL_2\} \qquad (7)$$

$B_2$ in Eq. (6) is the integration of $\epsilon_r \epsilon_0$ over axis $x$.

$$B_2 = \sum_i (\epsilon_r^{(i)} \epsilon_0)^{-1} d_i, i \in \{IL_1, HSO^1, Al_2O_3, HSO^2, IL_2\}, \qquad (8)$$

where $d_i$ denotes the thickness of the layer $i$. Equation (5) is further arranged using Eq. (2):

$$j(t_1 + \Delta t) = -(AR_{ESR} + B_2\Delta t)^{-1}\left[V_{ap}(t_1 + \Delta t) - V_c(t_1) - B_1\Delta t\right] \qquad (9)$$

Therefore, the total current density $j$ at the current time step $t_2 (= t_1 + \Delta t)$ can be calculated using Eq. (9) with the voltage across the SRMC at the previous time step ($V_c(t_1)$) and $B_1$ at the current time step $t_2$. The remaining task is to calculate $B_1$.

As such, the dc current density $j_{dc}$ considers electronic $j_e$ and ionic contributions $z_{Vo}qj_{dd}^{Vo}$: $j_{dc} = j_e + z_{Vo}qj_{dd}^{Vo}$, where $z_{Vo}$ and $q$ are the ionization number of an oxygen vacancy and elementary charge, respectively. Note that $j_{dd}^{Vo}$ is the flux of oxygen vacancies. Unlike the total current density $j$, the dc current density $j_{dc}$ varies along the layers because of the non-equilibrium distribution of oxygen vacancies at the applied voltage. This is due to the sluggish response of oxygen vacancies to the internal electric field because of their low diffusion coefficient $D_{Vo}$ (and thus mobility $M_{Vo}$). The flux $j_{dd}^{Vo}$ is driven by the gradient of electrochemical potential of oxygen vacancies, $\nabla \eta_{Vo}(= \nabla \mu_{Vo} + z_{Vo}q\nabla V)$. The chemical potential of dilute oxygen vacancies is denoted by $\mu_{Vo}(= \mu_{Vo}^0 + kT\ln(c_{Vo}/c_{Vo}^0))$, where $\mu_{Vo}^0$, $c_{Vo}^0$, $c_{Vo}$, $k$, and $T$ are the chemical potential and oxygen vacancy density at the reference state, oxygen vacancy density at a given state, Boltzmann constant, and lattice temperature, respectively. In the first-order approximation (FOA), the oxygen vacancy flux $j_{dd}^{Vo}$ is given by

$$j_{dd}^{Vo} = -q^{-1}c_{Vo}M_{Vo}\nabla \eta_{Vo} \qquad (10)$$

Considering the electrochemical potential gradient and Einstein relation, Eq. (10) becomes the celebrated drift-diffusion equation, $j_{dd}^{Vo} = -D_{Vo}\nabla c_{Vo} - z_{Vo}c_{Vo}M_{Vo}\nabla V$. However, the application of a high voltage to the SRMC of thin active layers (~4 nm) likely undermines the FOA, leading to the breakdown of FOA. In this case, we should consider the full velocity equation[44]:

$$v_{Vo} = \frac{D_{Vo}}{a_{Vo}}\left(e^{-\frac{a_{Vo}\nabla \eta_{Vo}}{2kT}} - e^{\frac{a_{Vo}\nabla \eta_{Vo}}{2kT}}\right), \qquad (11)$$

where $a_{Vo}$ is the hopping distance of an oxygen vacancy. Note that the electric field involved in $\nabla \eta_{Vo}$ in this equation is macroscopic electric field[45]. Equation (11) can further be arranged as

$$v_{Vo} = \frac{D_{Vo}}{a_{Vo}}e^{\frac{a_{Vo}}{2kT}(z_{Vo}q\nabla V + \nabla \mu_{Vo}^0 + kT\nabla c_{Vo})}\left(e^{-\frac{a_{Vo}}{kT}(z_{Vo}q\nabla V + \nabla \mu_{Vo}^0 + kT\nabla c_{Vo})} - 1\right) \qquad (12)$$

Here, we consider the gradient of reference chemical potential $\mu_{Vo}^0$ that can be ignored when considering ion migration within a homogeneous medium. However, the reference state chemical potential may vary along the multilayer in our SRMC. Eventually, the oxygen vacancy flux $j_{dd}^{Vo}(= c_{Vo}v_{Vo})$ can be calculated at all edges for given distributions of internal electrostatic potential and oxygen vacancy density. Inversely, the oxygen vacancy flux distribution determines the oxygen vacancy density according to the Fick's second law, $dc_i/dt = -\nabla j_{dd}^i$, so that the oxygen vacancy flux and density are in a self-consistent relation at a given distribution of internal electrostatic potential, which can be calculated iteratively. The Fick's second law was solved using the finite difference method, and the self-consistent relation was retained using the Newton-Raphson method. We used asymmetric boundary conditions such that the BE/HSO$^2$ interface forms blocking contact for oxygen vacancy while the TE serves as an oxygen vacancy reservoir by maintaining a particular oxygen vacancy density value.

As stated, we applied the quasi-static approximation to electronic distribution and dc current density regarding the large difference in diffusion coefficient (and thus mobility) between electron and oxygen vacancy. That is, the electronic distribution and current density are at equilibrium at any given time step, leading to space-invariant dc current density ($\nabla j_e = 0$) and the consequent time-invariant density ($dc_e/dt = 0$) at given distributions of internal electrostatic potential and oxygen vacancy density. For the simplicity of modeling, we used the following approximations: (i) both BE and TE conform to the free electron model with spherical Fermi surface, and (ii) both HSO$^1$ and HSO$^2$ have single conduction band minima. Given the experimentally observed thermally activated current, we considered the conduction band conduction of free electrons in the insulating trilayer structure:

$$j_e = nM_e\nabla \epsilon_f, \qquad (13)$$

where $n$, $M_e$, and $\epsilon_f$ denote free electron density, electron's mobility, and Fermi level (electron's electrochemical potential), respectively. The free electron density $n$ is calculated from the density of states of electrons $g(\epsilon) = 8\pi(2m_e^3)^{1/2}h^{-3}(\epsilon - \epsilon_f)^{1/2}$ and the Fermi-Dirac distribution function $f(\epsilon)$.

$$n = \int_{\epsilon_c}^{\infty} g(\epsilon)f(\epsilon)d\epsilon, \qquad (14)$$

where $\epsilon_c$ is the conduction band minimum. We assumed that the Al$_2$O$_3$ diffusion barrier is sufficiently thin to have negligible effects on the overall electron transport through the SRMC.

In the steady-state, the electronic current in Eq. (13) equals the interfacial electronic currents at bottom and top interfaces ($j_{e(B)}$ and $j_{e(T)}$, respectively), i.e., $j_{e(B)} = j_e = j_{e(T)}$. The bottom interfacial current $j_{e(B)}$ is the difference between electrode-to-insulator current ($j_{e(B)MI}$) and insulator-to-electrode current ($j_{e(B)IM}$), $j_{e(B)} = j_{e(B)MI} - j_{e(B)IM}$.

The current $j_{e(B)IM}$ is attributed to the electron transport from BE to insulator. Note that the current direction is opposite to the electron transport direction. Given that the experimental observations indicate the thermal activation of electron transport, we modeled $j_{e(B)IM}$ using the Schottky emission equation, which considers the Schottky barrier height (SBH) lowered by image force when the interfacial electric field $E_{int}$ is negative.

$$j_{e(B)IM} = -A^* T^2 e^{-q\phi_b/kT},$$

where $A^*$ means the modified Richardson constant. The SBH $\phi_b$ with image force is given by

$$\phi_b = \phi_b^0 - \left(\frac{q}{4\epsilon_r^{(i)}\epsilon_0}\max(0, -E_{int})\right)^{1/2},$$

where $\phi_b^0$ is the band offset (difference between the electron affinity of the insulator and work function of the cathode electrode). The function $\max(0, -E_{int})$ indicates a max function which is used to include the barrier lowering effect only when the interfacial field is negative. The opposite current $j_{e(B)MI}$ should be considered to keep a balance against $j_{e(B)IM}$. This current was modeled as $j_{e(B)MI} = qv_{MI}n_B$, where $v_{MI} = A^*T^2(qN_c)^{-1}$. The electron density at the bottom interface $n_B$ can be calculated using Eq. (14). The effective density of states of is denoted by $N_c$. Finally, we can calculate the electron current $j_{e(B)}(= j_{e(B)MI} - j_{e(B)IM})$. The same holds for the top interfacial current $j_{e(T)}$. Yet, the direction should be opposite; $j_{e(T)} = j_{e(T)IM} - j_{e(T)MI}$.

The algorithm for resistive switching dynamics calculation is detailed in the following pseudocode.

## Algorithm 1

Resistive switching dynamics.

**Input**: internal voltage profile $V$, oxygen vacancy profile $c_{Vo}$, total current density $j$, applied voltage $V_{ap}$ at time step $t$-1

**Output**: internal voltage profile $V$, oxygen vacancy profile $c_{Vo}$, total current density $j$ at time step $t$

**Parameters**: iteration error minima $\delta_{(l)0}$ and $\delta_{(m)0}$

Evaluate $j$, $j_{dc}$, $V$ at time step $t$ and applied voltage $V_{ap}$:

while $\delta_{(l)} > \delta_{(l)0}$ do

  calculate $j_e$ at given $V_{(l)}$

  while $\delta_{(m)} > \delta_{(m)0}$ do

    calculate $j_{dd(m)}$ at given $c_{Vo(m)}$, $V_{(l)}$

    calculate $c_{Vo\ (m+1)}$ using $j_{dd(m)}$ at given $V_{(l)}$ s.t. $\delta_{(m)}$ $(=|c_{Vo(m)} - c_{Vo(m+1)}|)$ decreases

$$\delta_{(m)} \leftarrow |c_{Vo\ (m)} - c_{Vo(m+1)}|$$

  calculate $j_{dc(l)}$ using $j_e$ and $j_{dd(m)}$ at given $V_{(l)}$

  calculate $V_{(l+1)}$ using $j_{dc(l)}$ s.t. $\delta_{(l)}$ $(=|V_{(l+1)} - V_{(l)}|)$ decreases

$$\delta_{(l)} \leftarrow |V_{(l+1)} - V_{(l)}|$$

  calculate $j_{(l+1)}$ using $j_{dc(l+1)}$ and $V_{(l+1)}$

$$j[t] \leftarrow j_{(l+1)}$$
$$j_{dc}[t] \leftarrow j_{dc(l+1)}$$
$$V[t] \leftarrow V_{(l+1)}$$

**Electrical Measurements of 160 × 160 and 320 × 320 CAs**. Unlike the 30 × 30 CA, the 160 × 160 and 320 × 320 CAs are not random-accessible because the number of row- and column-lines to be connected exceeds the number of currently available probes. Nevertheless, we programmed the whole CA to identify the effect of sneak current on the single predefined cell under measurement by tying all row-lines (common row-inhibit-line) except the signal row-line and all column-lines except (common column-inhibit-line) the signal column as shown in Supplementary Fig. 5. We attempted to program the unselected group 2 cells into their LRS by pulling up the common row-inhibit-line and pulling down the common column-inhibit-line. Likewise, an attempt to program the unselected group 1 into their LRS was made by pulling up the signal column-line (Common column-inhibit-line) while the common row-inhibit-line (Signal row-line) is grounded. However, the LRS could not directly be verified on individual cells because of the lack of random-accessibility. Instead, the success in set switching was indirectly verified such that the total current through the cells in each group was compared with the current extrapolated from a single cell. The measured value was comparable to the extrapolated value, indirectly identifying the success in programming.

The I–V measurement on the predefined selected cell was conducted by applying a stare-case voltage ($V_{op}$) to the signal column-line while (i) the signal row-line was grounded and (ii) stare-case inhibit voltages ($V_{inhibit1}$, $V_{inhibit2}$) were applied to the common row- and column-inhibit-lines, respectively. The step-wise changes of $V_{op}$, $V_{inhibit1}$, and $V_{inhibit2}$ were synchronized.

## Data availability

The data that support the findings of this study are available from the corresponding authors on reasonable request.

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

## Acknowledgements

G.H.K. would like to acknowledge a Korea Research Institute of Chemical Technology grant (Grant no. SS2021-20; Development of smart chemical materials for IoT devices). This work was partly supported by a research grant from the National Research Foundation of Korea under Grant no. NRF-2019R1C1C1009810. This research was also supported by the Ministry of Trade, Industry & Energy (grant number 20012002) and Korea Semiconductor Research Consortium program for the development of future semiconductor devices.

## Author contributions

K.J. performed the device fabrication and electrical characterization. J.K. and C.S. conducted the array characterization. J.J.R. and S.-J.Y. conducted the thin film deposition and chemical composition analysis. M.K.Y. presented the technical discussion with regard to the electrical and materials characteristics. D.S.J. performed the SRMC modeling and characterization. D.S.J. and G.H.K. supervised all experiments and compiled the manuscript.

## Competing interests

The authors declare no competing interests.
