## [Peer Review File · Nature Communications]

Reviewers' Comments:

Reviewer #1:

Remarks to the Author:

The manuscript reports on a passive, 100 kbit cross-bar array, made of self-rectifying non-volatile memory devices. The manuscript is well written and some of the reported metrics are appealing for in-memory computing. However, the manuscript seems to be developed as a technical report presenting device a variety of device metrics. This is also obvious even from the abstract. Therefore, I believe that the manuscript would be interesting for a more specialized journal focusing mostly on electronic engineering.

Reviewer #2:

Remarks to the Author:

In this work, the authors demonstrate a trilayer ($\text{Hf}_0.8\text{Si}_0.2\text{O}_2/\text{Al}_2\text{O}_3/\text{Hf}_0.5\text{Si}_0.5\text{O}_2$)-based self-rectifying resistive memory cell. They claim that the proposed memory cell satisfies the requirements for the main memory in memory-centric computing such as high selectivity, reliable two-bit operation, low read power, low read latency, excellent non-volatility, and complementary metal-oxide-semiconductor compatibility. In addition to the results of single memory cells, they also show the proposed memory cells can potentially operate in passive crossbar arrays. I think the work can eventually be published in Nature Communications. However, clear explanation of the switching mechanism and additional array data are still needed.

Comment 1. Device operations.

- a. In 'Resistive switching mechanism and current behavior of SRMC' section, the authors explain the resistive switching mechanism based on the barrier height change, which is caused by the oxygen vacancy redistribution, and claim that the improved retention characteristic is coming from the tri-layer structure of the proposed memory cell. However, Figure 3 b shows that both barrier heights of the top and bottom interfaces are reduced after the memory cell is switched from HRS to LRS. How can the set programming voltage cause the barrier height change in both interfaces by changing the oxygen vacancy profile? It may be better to show the expected oxygen vacancy profiles before and after the resistive switching. I
- b. If the asymmetry is caused by the different electrode materials as the authors claimed, then control experiments using different top and bottom electrode materials need to be performed to verify this hypothesis.
- c. According to the authors' explanation, a separate oxygen reservoir (HSO1) and an oxygen-blocking layer (Al_2O_3) are the main reason why the proposed memory cell has the improved retention. Why can the oxygen-blocking layer improve the retention? Is the oxygen migration between HSO1 and HSO2 the main source affecting the retention characteristic? Again, it will be better to provide the detailed explanation about the role of each layer and explain how blocking oxygen can enhance the retention characteristic.
- d. It seems the set operation was mostly reported at 85C. Is higher temperature programming required, for example, due to the oxygen blocking layer? Does room temperature switching require much higher programming voltage or become less reliable?

Comment 2. Array demonstration.

- a. As shown in Figure 5 and 6, showing the experimental results of the passive array with different size is a good way to demonstrate the ability of the proposed memory cells in the passive array operation. The authors claim that the big difference between the I-V curves of 160x160 array and 320x320 array is coming from the sneak current. Usually, the current through the sneak path is problematic when the unselected rows/columns are floated, but in this case the unselected rows/columns are biased so that the effect of sneak current will not be huge. Why does the 320x320 array show such different I-V curves depending on the schemes compared to 160x160 array? If parasitic components such as line resistance contributes to this difference, the authors

should clarify it through the array simulation. If it is related to the number of unselected cells, it should be discussed why the current increase (i.e. Scheme 4 in $320 \times 320 = 10 \times$ Scheme 4 in 160×160) is larger than the increment of the number of unselected cells (group 1 in $320 \times 320 = 2 \times$ group 1 in 160×160).

b. More importantly, the title highlights "100 kb passive crossbar array". However, for the 25kb and 100kb array, the authors seemed to have only measured 1 cell in the array (it was not clearly stated but implied in "we investigated the I-V behavior of a selected SRMC embedded in a 160×160 (~25 kb) and 320×320 (~100 kb) CA" in p13). This is useful but clearly insufficient. In fact, I am puzzled by how the authors programmed the arrays. The measurement was done by tying all the unselected cells together, which is reasonable since there will not be enough probes to connect all the 320 rows and 320 columns. However, if this is done through hardwiring, then how can the authors be sure that all these unselected cells were programmed to LRS (p13, 2nd paragraph)? In fact, I don't know how the authors can program these unselected cells to LRS at all. This was never discussed.

c. Related to the previous comment, instead of showing one single device I-V, the authors need to show the ability to program a pattern in the 1kb, 25kb, and 100kb arrays, ideally showing the yield and variations from all devices in the arrays.

Comment 3. Memory-centric application

a. According to the authors' argument, the main application of this system is the memory-centric computing. To efficiently perform a vector-matrix multiplication, which is the common operation in the memory-centric computing, the system may need to measure multiple memory cells at the same time rather than measure a single cell. Non-linear I-V characteristic, very small current level ($0.4 \sim 2$ nA) – which also leads to very long "read" time, and parasitic effects on the passive array may make the proposed system difficult to achieve an improved and reliable performance on the memory-centric computing application. It will be much better to show that a simple example of operations in memory-centric computing such as a vector-matrix multiplication can be done by using the proposed system or provide a system-level guideline for implementing the proposed system on the memory-centric computing.

b. If experimental study cannot be done in time, then an analysis of a memory-centric computing hardware system based on the devices and arrays will be useful. Even a simple analysis will be helpful for the readers.

c. Read disturb is an important issue, as the authors pointed out, however it was only tested for 10^9 reads. This seems insufficient.

d. Typo in p4, "however, its programming endurance is similar to that of dynamic RAM and static RAM". "similar" should be "much lower".

Reviewer #3:

Remarks to the Author:

The paper "Self-rectifying resistive memory in 100 kb passive crossbar arrays" by K. Jeon et al. describes a novel self-rectifying RRAM device, demonstrating a 320×320 array with 2-bit devices for inference purposes. The paper is well-structured, well-written and interesting. I have some questions and requests:

1- The device size is $2\mu\text{m} \times 2\mu\text{m}$, which is fairly large when thinking of high-density applications. In addition, the resistance of the device is quite high, around 1GOhm. Device scaling should be necessary, but, as authors demonstrated, resistance depends on the area, stronger scaling would lead to extremely high resistances, as also revealed by Supp. Fig. 3, reaching hundreds of GOhm. This would cause read issues, since no peripheral circuitry can detect such small currents in a short period of time. Is there any way to break this trade-off between area and resistance? I would just mention that, while this can be an issue for memory, it can be an advantage for inference, since there we read the current sum of multiple devices (this discussion ties with request 4).

2- Authors showed good results for data retention after baking at LRS (Fig. 1d). I would also add HRS data retention after baking.

3- Authors reveal 100 kb crossbar array, however the amount of data taken from such array is extremely little. I would be curious to see the distribution of the 4 levels (2 bits) plus a retention measurement over the entire array.

4- Authors introduce such novel crossbar array for inference. It would be very interesting to perform a real inference activating all rows and reading the corresponding current through columns. In particular, it would be very interesting to see the inference results programming all devices in HRS, or first level LRS, or second or third, and check that the corresponding aggregate current increases accordingly. Even more complicated combinations of the four levels could be implemented, but I leave to authors whatever works best.

Responses to reviewers' comments

Manuscript #: NCOMMS-20-27954

Title: Self-rectifying resistive memory in 100 kb passive crossbar arrays

Authors: Kanghyeok Jeon, Jeeson Kim, Jin Joo Ryu, Seung-Jong Yoo, Choongseok Song, Min Kyu Yang, Doo Seok Jeong, Gun Hwan Kim

We greatly acknowledge the valuable comments on our manuscript from the reviewers. The comments and suggestions indeed helped us improve the manuscript. Based on the reviewers' comments and suggestions, we revised our manuscript thoroughly. The revision made is highlighted using blue color.

Reviewer #1 (Remarks to the Author):

The manuscript reports on a passive, 100 kbit cross-bar array, made of self-rectifying non-volatile memory devices. The manuscript is well written and some of the reported metrics are appealing for in-memory computing. However, the manuscript seems to be developed as a technical report presenting device a variety of device metrics. This is also obvious even from the abstract. Therefore, I believe that the manuscript would be interesting for a more specialized journal focusing mostly on electronic engineering.

Answer: We agree on the reviewer's view. We significantly revised the manuscript to highlight the scientific contribution of our study with regard to the mechanism for the switching in our SRMC. To this end, we modeled the switching behavior using Maxwell's fourth equation, which enabled us to calculate the current (electronic dc current + ionic dc current + displacement current) through the SRMC in time and voltage domains. In this modeling framework, we could simulate the current response to quasi-static stair-case voltages and voltage pulses as well as memory retention. Besides, the simulation yielded the oxygen vacancy dynamics in space and time domains at given voltages, which offers insight into the experimental observations from a scientific viewpoint. We hope that the balance between the technical and scientific contributions of our study is eligible for the publication in Nature Communications.

Author action: We added a new subsection "Modeling of switching in SRMCs" in the Results section. The detail of the modeling is addressed in "Modeling of resistive switching dynamics" in the Methods

section.

Reviewer #2 (Remarks to the Author):

In this work, the authors demonstrate a trilayer ($\text{Hf}_{0.8}\text{Si}_{0.2}\text{O}_2/\text{Al}_2\text{O}_3/\text{Hf}_{0.5}\text{Si}_{0.5}\text{O}_2$)-based self-rectifying resistive memory cell. They claim that the proposed memory cell satisfies the requirements for the main memory in memory-centric computing such as high selectivity, reliable two-bit operation, low read power, low read latency, excellent non-volatility, and complementary metal-oxide-semiconductor compatibility. In addition to the results of single memory cells, they also show the proposed memory cells can potentially operate in passive crossbar arrays. I think the work can eventually be published in Nature Communications. However, clear explanation of the switching mechanism and additional array data are still needed.

Answer: We thank the reviewer for the good word. We did our best to address the reviewer's concerns.

Comment 1. Device operations.

- a. In 'Resistive switching mechanism and current behavior of SRMC' section, the authors explain the resistive switching mechanism based on the barrier height change, which is caused by the oxygen vacancy redistribution, and claim that the improved retention characteristic is coming from the tri-layer structure of the proposed memory cell. However, Figure 3 b shows that both barrier heights of the top and bottom interfaces are reduced after the memory cell is switched from HRS to LRS. How can the set programming voltage cause the barrier height change in both interfaces by changing the oxygen vacancy profile? It may be better to show the expected oxygen vacancy profiles before and after the resistive switching.

Answer: We thank the reviewer for the critical comment. The SRMC is viewed as a back-to-back Schottky diode with asymmetric Schottky barrier heights. If the thickness of the active layer (here the trilayer) was larger than two times the characteristic Debye length, the change of oxygen vacancy density (working as space charge) in the vicinity of one interface would not alter the macroscopic electric field distribution in the vicinity of the other interface. Given the space charge density considered in this study ($<10^{21} \text{ cm}^{-3}$), the Debye length is larger than the thickness of our trilayer. Therefore, in our SRMC, the change of vacancy density near one interface definitely perturbs the field distribution near the other interface. Regarding the Poisson's equation, it is deduced that interfacial electric field strength at both interfaces is proportional to the vacancy density. The detailed role of the interfacial electric field differs for different interfacial conduction mechanisms. Nevertheless, it is common to consider the interfacial electric field to reduce the Schottky barrier height. Therefore, we expect that the change in the vacancy density within the trilayer alters the both Schottky barrier

heights such that the larger the vacancy density, the lower the Schottky barrier height at both interface.

Author action: Following the reviewer’s comment, we modeled the resistive switching behavior of our SRMC from scratch. The model used Maxwell’s fourth equation applied to discretized one-dimensional SRMC, which yielded the current (electronic dc current + ionic dc current + displacement current) through the SRMC in time and voltage domains. Simultaneously, we could acquire the oxygen vacancy dynamics in time and space domains at given voltages. The results are addressed in a new section “Modeling of switching in SRMCs” in the Results section. The calculation in detail is explained in the Methods section.

- b. If the asymmetry is caused by the different electrode materials as the authors claimed, then control experiments using different top and bottom electrode materials need to be performed to verify this hypothesis.

Answer: We would like to point that we did not claim that the use of asymmetric electrodes is the origin of the asymmetric I-V. Because we are aware that often the symmetric electrode causes asymmetric I-V (e.g., Nanoscale 5, 6363, 2013), we wanted to avoid any assured statements. Instead, we speculate feasible contributions to the asymmetry, and our use of asymmetric electrodes can be a reason for the asymmetry in I-V. We modified the statement on Page 8 to make even less bold.

Nevertheless, we in fact attempted to use different electrodes during our device structure optimization process. The Ru and TiN combination turned out to be the best combination. For instance, W/TiN and Ni/TiN combinations failed to reproduce the reliable self-rectifying behavior of the Ru/TiN combination. Please see the figure below (Fig. R1).

Figure R1. Top electrode dependent resistive switching characteristics (W and Ni)

Author action: The statement “the largely asymmetric I - V behavior is likely due to the use of asymmetric metal electrodes and thus asymmetric interfacial barrier heights” on Page 8 was changed to “the largely asymmetric I - V behavior may be due to the use of asymmetric metal electrodes and thus asymmetric interfacial barrier heights” on the same page.

- c. According to the authors’ explanation, a separate oxygen reservoir (HSO¹) and an oxygen-blocking layer (Al₂O₃) are the main reason why the proposed memory cell has the improved retention. Why can the oxygen-blocking layer improve the retention? Is the oxygen migration between HSO¹ and HSO² the main source affecting the retention characteristic? Again, it will be better to provide the detailed explanation about the role of each layer and explain how blocking oxygen can enhance the retention characteristic.

Answer: Throughout device modeling, we found the following three indications.

- Indication (i): The lower vacancy density in HSO² than HSO¹ in both resistance states.
- Indication (ii): small change in vacancy density in both states over time, i.e., small $c_{V_o,t=0} - c_{V_o,t=7200}$ in both layers. c_{v_o} denotes the oxygen vacancy density.
- Indication (iii): small difference in vacancy density between LRS and HRS, i.e., small $\Delta c_{V_o,t=0}$ in HSO².

Regarding Indication (i), the resistance state of our SRMC may be mainly dictated by the oxygen vacancy density in HSO¹ rather than HSO². This is because the interfacial electric fields $E(0)$ and $E(d)$ of the bottom and top interface, respectively, are mainly determined by large space charge density as best explained using the following equation.

$$\begin{cases} E(0) = -\frac{V_{ap}}{d} - \frac{q}{\epsilon_r \epsilon_0 d} \int_0^d \int_0^x c_{V_o}(x') dx' dx \\ E(d) = -\frac{V_{ap}}{d} + \frac{q}{\epsilon_r \epsilon_0} \int_0^d \left(c_{V_o}(x') - \frac{1}{d} \int_0^x c_{V_o}(x') dx' \right) dx \end{cases} \quad \text{Eq. (R1)}$$

where V_{ap} , d , ϵ_r , and ϵ_r are the applied voltage, trilayer thickness, dielectric constant, and vacuum permittivity, respectively. In this equation, integrating the oxygen vacancy density over HSO² is much smaller than over HSO¹ because the larger vacancy density HSO¹ in than HSO².

The indication (ii) is the direct cause of the excellent LRS retention.

The indication (iii) is caused by the Al₂O₃ oxygen-blocking layer. The low diffusion coefficient of Al₂O₃ hinders oxygen vacancies from entering into (leaving from) HSO² during set (reset) switching.

This can be seen in comparison with the $\text{HSO}^1/\text{HSO}^2$ cell whose oxygen vacancy distributions in both states are plotted in Fig. R2. Figure R2f identifies that the lack of the oxygen-blocking layer allows a large number of oxygen vacancies to enter into (leave from) HSO^2 , unlike the trilayer. Thus, the role of the Al_2O_3 oxygen-blocking layer in switching is to confine the active switching region to HSO^1 . The better LRS retention for the trilayer than $\text{HSO}^1/\text{HSO}^2$ is understood in terms of the confined switching to HSO^1 . According to Eq. (R1), the larger the oxygen vacancy density, the larger the interfacial electric field evolves; the larger electric field in the vicinity of the top interface $E(d)$ drives the more oxygen vacancies back to the source (Ru). The unconfined cell ($\text{HSO}^1/\text{HSO}^2$) holds the more oxygen vacancies over than the trilayer, and the larger $E(d)$ releases the more oxygen vacancies to the vacancy source. To show this, we evaluated the areal density of oxygen vacancies in each layer for the trilayer and $\text{HSO}^1/\text{HSO}^2$ bilayer. The areal density was calculated by integrating the vacancy density over the HSO^1 or HSO^2 region. The results are shown in Fig. R2g, which identifies the larger decay in oxygen vacancy density in HSO^2 in the bilayer than the trilayer.

Figure R2. **a** One-dimensional configuration of the SRMC for simulation. **b** Simulated I - V loop (quasi-static behavior) in comparison with experimental data. **c** Simulated switching behaviors in response to voltage pulses of different widths and amplitudes. **d** Simulated LRS retention for the HSO^2 -only cell (Dev 1), $\text{HSO}^1/\text{HSO}^2$ cell (Dev 2), and $\text{HSO}^1/\text{Al}_2\text{O}_3/\text{HSO}^2$ SRMC (Dev 3). **e, f**

Simulated oxygen vacancy distributions in the trilayer SRMC and HSO¹/HSO² cell in the LRS (upper panel) and HRS (lower panel). The change of the distribution in each state was monitored in the time range (0 – 7200 s). **g** Retention of areal density of oxygen vacancies in the LRS in each layer of the trilayer and bilayer cells.

Author action: These new results are addressed in a new section “Modeling of switching in SRMCs” in the Results section. The calculation in detail is explained in the Methods section.

d. It seems the set operation was mostly reported at 85C. Is higher temperature programming required, for example, due to the oxygen blocking layer? Does room temperature switching require much higher programming voltage or become less reliable?

Answer: We chose a memory operation temperature of 85 °C because this temperature is the upper bound of the industrial temperature range (-40 – 85 °C). We rather exposed our SRMCs to harsh environments. In fact, we do not see any significant effects of temperature on switching operation other than the overall decrease of device current, which is due to the thermally activated current transport in nature in our cells. This is another advantage of our SRMCs. Reviewer can see the quasi-static I-V behaviors at temperatures below 85 °C in Fig. R3.

Figure R3. Quasi-static I-V behaviors at various temperatures

Author action: We added the following statement in the Results section on Page 6.

“We chose a memory operation temperature of 85 °C which is the upper bound of the industrial temperature range (-40 – 85 °C).”

Comment 2. Array demonstration.

a. As shown in Figure 5 and 6, showing the experimental results of the passive array with different size is a good way to demonstrate the ability of the proposed memory cells in the passive array operation. The authors claim that the big difference between the I-V curves of 160x160 array and 320x320 array is coming from the sneak current. Usually, the current through the sneak path is problematic when the unselected rows/columns are floated, but in this case the unselected rows/columns are biased so that the effect of sneak current will not be huge. Why does the 320x320 array show such different I-V curves depending on the schemes compared to 160x160 array? If parasitic components such as line resistance contributes to this difference, the authors should clarify it thorough the array simulation. If it is related to the number of unselected cells, it should be discussed why the current increase (i.e. Scheme 4 in 320x320 = 10 x Scheme 4 in 160x160) is larger than the increment of the number of unselected cells (group 1 in 320x320 = 2 x group 1 in 160x160).

Answer: Thank the reviewer for the fruitful comment that brought flaws to our attention. Certainly, we can ignore the line resistance effect because the total line resistance of the 320×320 array is approximately $26 \text{ k}\Omega$ —much lower than the SRMC resistance by more than three orders of magnitude over the entire voltage range. Please see the measured I-V on the line-only, i.e., without SRMCs, 320×320 array in Fig. R4.

Figure R4. Experimentally measured line resistance

The reviewer's estimation of the effect of the unselected cells on the current measure definitely makes sense to us, which we could not recognize before. Although the number of unselected cells in group 2 is proportional to N^2 (N : the number of cells in one signal line), the current level of each unselected

cell is insufficient (ca. 1 pA/cell) to dictate the total leakage current level. Therefore, we concluded that unexpected defects were incorporated into our previous 320×320 CA.

We fabricated a new 320×320 CA and measured the quasi-static I-V curves for different Schemes, which are plotted in Fig. R4f.

Author action: We fabricated a new 320 × 320 CA and measured the resistive switching characteristics. Accordingly, we replaced the previous Fig. 6f with the new data plot (Fig. R5f).

Figure R5: 160 × 160 and 320 × 320 CAs of SRMCs. **a–d** Illustrations of Schemes 1–4 and voltage across different cells indicated by different colors. I-V loops of selected cell that was embedded in **e** 160 × 160 and **f** 320 × 320 CA.

b. More importantly, the title highlights “100 kb passive crossbar array”. However, for the 25kb and 100kb array, the authors seemed to have only measured 1 cell in the array (it was not clearly stated but implied in “we investigated the I-V behavior of a selected SRMC embedded in a 160 × 160 (~25 kb) and 320 × 320 (~100 kb) CA” in p13). This is useful but clearly insufficient. In fact, I am puzzled by how the authors programmed the arrays. The measurement was done by tying all the unselected cells together, which is reasonable since there will not be enough probes to connect all the 320 rows and 320 columns. However, if this is done through hardwiring, then how can the authors be sure that all these unselected cells were programmed to LRS (p13, 2nd paragraph)? In fact, I don’t know how the

authors can program these unselected cells to LRS at all. This was never discussed.

Answer: We thank the reviewer for the comment. Indeed, we did not elaborate the method for the large arrays. We detailed the method in the revised manuscript. The reviewer is right. The larger arrays (160×160 and 320×320) are not random accessible; instead, we predefine a single (hardwired) cell subject to switching measurement while the other cells (unselected group 1 and 2 cells) are subject to a voltage through common leads. The measurement configuration is shown below.

Figure R6. Measurement configuration for the 160×160 and 320×320 CAs

The unselected group 2 cells were programmed to the LRS by pulling down the common row inhibit line and applying a voltage to the common column inhibit line. We agree with the reviewer on the point that the success in switching to LRS cannot be identified because of lack of random accessibility. Nevertheless, we could indirectly identify it by comparing the measured current through parallel cells in group 2 and the current extrapolated from a single cell. The two values turned out to be comparable, so that we estimate the success in switching. Nevertheless, the reviewer is right that we are not sure the true conductance distribution after the switching, and thus it is not sure if the conductance distribution is the worst case. In the revised manuscript, we elaborate the measurement of the large CAs in the Methods section as well as the main text. Additionally, we refer to the conductance distribution as ‘seemingly’ worst case since it is not sure.

Author action: We revised the main text on Pages 17 and 18 as follows.

“To examine the sneak current from the unselected groups 1 and 2 cells, we attempted to program all unselected groups 1 and 2 cells into their LRS and subsequently examined the I - V characteristics of the selected cell. The measured I - V loops for the selected cell embedded in the 160×160 CA are shown in Fig. 7e. The different biasing schemes caused a negligible difference

in the I - V loops of the selected cell. This confirms that the self-rectifying and nonlinear I - V behavior of the SRMC maintains a sufficiently low current through the unselected cells to enable the true current to be read through the selected cell.

The 320×320 CA allows the sneak current to vary the ground line current more obviously than the smaller CAs, yielding more obviously distinct I - V loops depending on the voltage-application scheme (Fig. 7f). Scheme 2 yields a lower current than Scheme 1 over the whole voltage range, whereas the largest current level was yielded by Schemes 3 and 4. This is because Scheme 2 applies the lowest voltage to unselected group 1 cells, which share the same row-line as the selected cell. Nevertheless, the switching behavior of the selected cell indicates two distinct states despite the sneak current in this ‘seemingly’ worst-case conductance distribution.”

Additionally, we added a subsection dedicated to the measurement method for the large CAs to the Methods section on Pages 28 and 29 as follows.

“Electrical Measurements of 160×160 and 320×320 CAs

Unlike the 30×30 CA, the 160×160 and 320×320 CAs are not random-accessible because the number of row- and column-lines to be connected exceeds the number of currently available probes. Nevertheless, we programmed the whole CA to identify the effect of sneak current on the single predefined cell under measurement by tying all row-lines (common row-inhibit-line) except the signal row-line and all column-lines except (common column-inhibit-line) the signal column as shown in Fig. 10. We attempted to program the unselected group 2 cells into their LRS by pulling up the common row-inhibit-line and pulling down the common column-inhibit-line. Likewise, an attempt to program the unselected group 1 into their LRS was made by pulling up the signal column-line (Common column-inhibit-line) while the common row-inhibit-line (Signal row-line) is grounded. However, the LRS could not directly be verified on individual cells because of the lack of random-accessibility. Instead, the success in set switching was indirectly verified such that the total current through the cells in each group was compared with the current extrapolated from a single cell. The measured value was comparable to the extrapolated value, indirectly identifying the success in programming.

The I - V measurement on the predefined selected cell was conducted by applying a stare-case voltage (V_{op}) to the signal column-line while (i) the signal row-line was grounded and (ii) stare-case inhibit voltages ($V_{inhibit1}$, $V_{inhibit2}$) were applied to the common row- and column-inhibit-lines, respectively. The step-wise changes of V_{op} , $V_{inhibit1}$, and $V_{inhibit2}$ were synchronized.”

c. Related to the previous comment, instead of showing one single device I - V , the authors need to

show the ability to program a pattern in the 1kb, 25kb, and 100kb arrays, ideally showing the yield and variations from all devices in the arrays.

Answer: We agree with the reviewer. Following the reviewer’s comment, we conducted array-level experiments on column-wise summed current production using the 30×30 CA which is random accessible. We will explain the experiments in detail when addressing the reviewer’s comment 3. To be honest, for the moment it is not possible to address individual cells in the 160×160 and 320×320 CAs. To do so, we need 80 and 160 commercially available four-channel analog switches; placing them on a single measurement board is a fairly daunting task. Alternatively, we could fabricate FEOL peripheral circuits including column and row line decoders and switch matrices, which is co-integrated with our CAs. However, the limited reliability of large scale CA fabrication processes at the lab scale makes us hesitate to attempt. Hopefully, the reviewer understands the situation.

Nevertheless, surely, the large CAs does not include any short-circuited cells, which would otherwise clearly show large current in the I - V measurements. Although we cannot remark the presence of open-circuited cells in the CAs, the lack of short-circuited cells may indicate the high reliability of the CAs.

Answer action: We added a new section “Vector-matrix multiplication acceleration using the 30×30 crossbar array” in the Result section. We will give the detail of this section when addressing the reviewer’s comment 3.

Comment 3. Memory-centric application

- a. According to the authors’ argument, the main application of this system is the memory-centric computing. To efficiently perform a vector-matrix multiplication, which is the common operation in the memory-centric computing, the system may need to measure multiple memory cells at the same time rather than measure a single cell. Non-linear I - V characteristic, very small current level ($0.4 \sim 2$ nA) – which also leads to very long “read” time, and parasitic effects on the passive array may make the proposed system difficult to achieve an improved and reliable performance on the memory-centric computing application. It will be much better to show that a simple example of operations in memory-centric computing such as a vector-matrix multiplication can be done by using the proposed system or provide a system-level guideline for implementing the proposed system on the memory-centric computing.
- b. If experimental study cannot be done in time, then an analysis of a memory-centric computing hardware system based on the devices and arrays will be useful. Even a simple analysis will be

helpful for the readers.

Answer: We thank the reviewer for the valuable suggestion. We conducted a new set of experiments to identify the feasible VMM ($\mathbf{w} \times \mathbf{x}$; $\mathbf{w} \in \mathbb{Z}^{30 \times 30}$, $\mathbf{x} \in \mathbb{Z}^{30}$) acceleration using our 30×30 CA by means of column-wise dot product. We chose four random matrices ($\mathbf{w}_1 - \mathbf{w}_4$) filled with 2-bit integers. The matrices differ in sparsities (0, 25, 51, 55%). First, each matrix was mapped onto a 30×30 CA by programming each individual cell using Scheme 2. The elements $x[i]$ in the vector \mathbf{x} were 1-bit values; ‘0’ and ‘1’ were encoded as 0 V and 2 V to the row lines. The intermediate result of the VMM is the analog current vector \mathbf{j} measured at the column lines. We measured this intermediate result for the four matrices and compared them with the results extrapolated from the individual cells. The comparison clearly indicates no parasitic effect on the summed current.

The results are shown in the figure below.

Figure R7. Acceleration of vector-matrix multiplication using the 30×30 CA

Additionally, we explain the key considerations when processing the intermediate analog product. They include energy consumption and quantization bit-width. A common approach employs current-to-voltage converters and ADCs to output the final VMM. Yet, one should pay attention to the

considerable power consumed by the ADCs, which likely dominates the total power of the RRAM-based VMM acceleration systems. The quantization bit-width is the key to the performance. When applied to inference acceleration, the limited quantization bit-width can degrade the inference accuracy more significantly than the limited quantization bit-width of weights as shown in recent neural networks such as DoReFa-Net.

Moreover, we provide a VMM acceleration scheme for multibit vector x . Because of the nonlinear I-V characteristics of our SRMC, we cannot encode the element value as the height of the applied voltage. Instead, a suitable way is to encode the multibit element $x[i]$ using time-division multiplexing. Although this scheme comes at the cost of an additional delay proportional to the bit-width of $x[i]$, it may support the high reliability of multiplication. The scheme is shown in the figure below.

Figure R8. Acceleration of multibit vector-matrix multiplication

Author action: We added a new section “Vector-matrix multiplication acceleration using the 30×30 crossbar array” attached below.

“Vector-matrix multiplication acceleration using the 30×30 crossbar array

Finally, we identified the feasible acceleration of vector-matrix multiplication ($w \times x$; $w \in \mathbb{Z}^{30 \times 30}$, $x \in \mathbb{Z}^{30}$) by reducing the computational complexity to $O(n)$. To this end, we aimed to calculate a dot product $w[i, :] \cdot x$ at one cycle, where $w[i, :]$ denotes the i th row of matrix w . We restricted the elements w of matrix w to 2-bit integers ($w \in \{0,1,2,3\}$) and the elements x to 1-bit integers ($x \in \{0,1\}$). The matrix w was transposed and mapped onto our 30×30 SRMC CA (conductance of each cell $\in \{HRS, L1, L2, L3\}$). The vector x with 1-bit integer elements was

encoded as a voltage array V_{ap} ($V_{ap} \in \{0, 2V\}$) and applied to the 30 row-lines of the CA (Fig. 8a). The current measured at the i th column-line was the intermediate result of the dot product $\mathbf{w}[i, :] \cdot \mathbf{x}$. As depicted in Fig. 8b, we addressed one column at one cycle by pulling down the chosen column-line to the ground while inhibit voltages ($V_{inhibit}$) were applied to the rest of column-lines ($2/3V_{ap}$), so that we reduced the complexity to $O(n)$, which is otherwise $O(n^2)$.

We chose four random matrices (\mathbf{w}_1 , \mathbf{w}_2 , \mathbf{w}_3 , and \mathbf{w}_4) of different sparsities (0, 25, 51, and 55%, respectively). The percentage of each integer (0, 1, 2, 3) in each matrix is shown in Fig. 8c. The chosen matrices were mapped onto four 30×30 CAs such that the individual cells of the CAs were randomly accessed and programmed to the correct conductance states using Scheme 2. The programmed conductance map for each matrix is shown in Fig. 8d – g. The conductance of each SRMC was individually read out at a read-out voltage of 2 V to acquire the maps. We then performed the dot product $\mathbf{w}[i, :] \cdot \mathbf{x}$ for each i at one cycle with vector \mathbf{x} of ones, i.e., $\mathbf{x} = [1, 1, \dots, 1]$. The vector-matrix multiplication operation for each matrix thus consumes 30 column-line-addressing cycles, yielding a current vector \mathbf{j} ($\in \mathbb{R}^{30}$) as the intermediate product (Fig. 8d – g). The measured current at each column-line is almost identical to the current value extrapolated from each cell current in the same column, indicating marginal disturbance from the unselected cells. For the multiplication with four matrices (\mathbf{w}_1 , \mathbf{w}_2 , \mathbf{w}_3 , and \mathbf{w}_4), the CA domain consumes powers of 4.22, 3.44, 2.83, and 2.69 $\square W$, respectively. The considered multiplication is the worst case in terms of power consumption because of the extremely dense vector \mathbf{x} (of ones).

To output the final product \mathbf{z} ($\mathbf{z} = \mathbf{w} \times \mathbf{x}$; $\mathbf{z} \in \mathbb{Z}^{30}$), current from the i th column j_i for all i needs to be encoded as a binary number, which subsequently enters into the near-memory digital domain for additional processing. A common method is to convert the summed current to voltage and subsequently to quantize the converted voltage using an analog-to-digital converter (ADC)⁵⁰. Alternatively, the summed current can directly be converted to a binary value using a current sense amplifier (CSA) with multiple reference currents that are iteratively compared with the summed current⁵¹. In either way, the important consideration is twofold: (i) energy consumption and (ii) bit-width of the product \mathbf{z} . Regarding power consumption, ADCs are well known to consume a considerable amount of energy insomuch as the total energy consumption of an RRAM-based inference accelerator is dominated by the ADCs¹³. An alternative method using a CSA⁵¹ keeps the static current from the chosen line flowing while the summed current being converted iteratively, causing additional energy consumption. The bit-width should be chosen carefully to avoid the performance, i.e., inference, degradation by the quantization bit-width. As shown in quantized neural networks such as DoReFa-Net⁵², the resolution, i.e., bit-width, of activations more critically determines the inference accuracy than that of weights. The activation resolution is dictated by the

bit-width of the output z . Therefore, the bit-width of the product z is an important consideration in the design of summed current-encoding circuits.

Regarding multibit factor \mathbf{x} , time-division multiplexing is a desirable method by encoding the vector \mathbf{x} as shown in Fig. 9. Because of the nonlinear I - V behavior in the LRS of our SRMCs, encoding a factor as input voltage amplitude is unsuitable unlike linear I - V cases^{14, 53}. The l -bit elements $x[i]$ are time-division multiplexed from the least significant bits (LSBs) to the most significant bits (MSBs) and are applied to the row-lines at one column-line addressing cycle for the dot product $\mathbf{w}[i, :] \cdot \mathbf{x}$. Thus, each dot product cycle includes l sub-cycles. The output current at each sub-cycle is encoded as a binary value and subsequently multiplied by 2^{k-1} , where k denotes the digit corresponding to the sub-cycle. The results are finally summed to output the dot product $\mathbf{w}[i, :] \cdot \mathbf{x}$.”

- c. Read disturb is an important issue, as the authors pointed out, however it was only tested for 10e9 reads. This seems insufficient.

Answer: Following the reviewer’s comment, we tested the read operations up to 10^{10} times and revised Figure 1.

Author action: We revised Figure 1f. The revised figure is shown below.

Figure R9. Read disturb characteristic up to 10^{10} times

- d. Typo in p4, “however, its programming endurance is similar to that of dynamic RAM and static RAM”. “similar” should be “much lower”.

Answer: Thank the reviewer for the comment.

Author action: We corrected the typo as follows.

“however, its programming endurance is much lower than dynamic RAM and static RAM.”

Reviewer #3 (Remarks to the Author):

The paper “Self-rectifying resistive memory in 100 kb passive crossbar arrays” by K. Jeon et al. describes a novel self-rectifying RRAM device, demonstrating a 320x320 array with 2-bit devices for inference purposes. The paper is well-structured, well-written and interesting. I have some questions and requests:

- 1- The device size is 2umx2um, which is fairly large when thinking of high-density applications. In addition, the resistance of the device is quite high, around 1GOhm. Device scaling should be necessary, but, as authors demonstrated, resistance depends on the area, stronger scaling would lead to extremely high resistances, as also revealed by Supp. Fig. 3, reaching hundreds of GOhm. This would cause read issues, since no peripheral circuitry can detect such small currents in a short period of time. Is there any way to break this trade-off between area and resistance? I would just mention that, while this can be an issue for memory, it can be an advantage for inference, since there we read the current sum of multiple devices (this discussion ties with request 4).

Answer: We thank the reviewer for the insightful comment. The reviewer is right. If we use our SRMCs as random access memory cells, we cannot use current sense amplifiers (CSAs) because of such a low current level. Instead, we can use a voltage sensing scheme, but that causes a long read-out delay. However, as the reviewer pointed out, such high resistance can be beneficial in terms of power consumption; each cell allows very low static current to flow. It turned out that the CA domain consumes only a few μW for vector-matrix multiplication even in the worst case due to the very dense vector. We will address this point when addressing the reviewer’s request 4.

Nevertheless, even if large matrices and vectors are considered, when they are sparse, the summed current level is low which again causes a long read-out delay. In fact, we aim to use our SRMC arrays to accelerate the inference of XNOR-Net which is a type of binarized neural network with binary weights and binary activations. The XNOR operation needs a pair of resistive memory cells to represent ‘-1’ and ‘1’. In this case, when the activation is ‘1’, the summed current is always larger than $I_{\text{LRS}} \times$ (half the cells in one column). Thus, our high resistance SRMC is one of the perfect candidates for the applications. Normal resistive memory cells with low resistance in their LRS consume too much power in this case.

- 2- Authors showed good results for data retention after baking at LRS (Fig. 1d). I would also add HRS data retention after baking.

Answer: Following the reviewer’s comment, we measured the HRS data retention after baking under

the same condition as the LRS. We found a negligible change in resistance as shown in **Fig. R10**.

Author action: We added the newly measured HRS retention data to Fig. 1 (Fig. 1d) in the revised manuscript.

Figure R10. Statistics of HRS data retention

3- Authors reveal 100 kb crossbar array, however the amount of data taken from such array is extremely little. I would be curious to see the distribution of the 4 levels (2 bits) plus a retention measurement over the entire array.

Answer: We thank the reviewer for the valuable comment. As described in our original manuscript, only the 30×30 array is fully addressable while the other 25 kb and 100 kb is not because of insufficient number of probe. Hopefully, the reviewer understands the measurement capacity at the lab scale. We highlight the 100 kb array because our SRMC supports good readability in the 100 kb array. Instead, following the reviewer's comment, we measured the 2-bit operation and retention characteristics of the 30×30 CA (Fig. R11).

Figure R11. Two-bit operation and retention characteristics of the 30×30 CA.

Additionally, we performed the two-bit operation of the predefined cell in the 100 kb CA (Fig. R12). The results highlight reliable two-bit operation in the 100 kb CA.

Figure R12. Two-bit operation in the 320×320 CA.

Author action: We added the two-bit operation results from the 30×30 CA and 320×320 CA to the Supplementary Information.

- 4- Authors introduce such novel crossbar array for inference. It would be very interesting to perform a real inference activating all rows and reading the corresponding current through columns. In particular, it would be very interesting to see the inference results programming all devices in HRS, or first level LRS, or second or third, and check that the corresponding aggregate current increases accordingly. Even more complicated combinations of the four levels could be implemented, but I leave to authors whatever works best.

Answer: We thank the reviewer for the valuable comment. We conducted a new set of experiments to identify the feasible VMM ($\mathbf{w} \times \mathbf{x}$; $\mathbf{w} \in \mathbb{Z}^{30 \times 30}$, $\mathbf{x} \in \mathbb{Z}^{30}$) acceleration using our 30×30 CA by means of column-wise dot product. We chose four random matrices ($\mathbf{w}_1 - \mathbf{w}_4$) filled with 2-bit integers. The matrices differ in sparsities (0, 25, 51, 55%). First, each matrix was mapped onto a 30×30 CA by programming each individual cell using Scheme 2. The elements $x[i]$ in the vector \mathbf{x} were 1-bit values; '0' and '1' were encoded as 0 V and 2 V to the row lines. The intermediate result of the VMM is the analog current vector \mathbf{j} measured at the column lines. We measured this intermediate result for the four matrices and compared them with the results extrapolated from the individual cells. The comparison clearly indicates no parasitic effect on the summed current.

The results are shown in the figure below.

Figure R13. Acceleration of vector-matrix multiplication using the 30×30 CA

Additionally, we explain the key considerations when processing the intermediate analog product. They include energy consumption and quantization bit-width. A common approach employs current-to-voltage converters and ADCs to output the final VMM. Yet, one should pay attention to the considerable power consumed by the ADCs, which likely dominates the total power of the RRAM-based VMM acceleration systems. The quantization bit-width is the key to the performance. When applied to inference acceleration, the limited quantization bit-width can degrade the inference accuracy more significantly than the limited quantization bit-width of weights as shown in recent neural networks such as DoReFa-Net.

Moreover, we provide a VMM acceleration scheme for multibit vector x . Because of the nonlinear I-V characteristics of our SRMC, we cannot encode the element value as the height of the applied voltage. Instead, a suitable way is to encode the multibit element $x[i]$ using time-division multiplexing. Although this scheme comes at the cost of an additional delay proportional to the bit-width of $x[i]$, it may support the high reliability of multiplication. The scheme is shown in the figure below.

Figure R14. Acceleration of multibit vector-matrix multiplication

Author action: We added a new section “Vector-matrix multiplication acceleration using the 30×30 crossbar array” attached below.

“Vector-matrix multiplication acceleration using the 30×30 crossbar array

Finally, we identified the feasible acceleration of vector-matrix multiplication ($\mathbf{w} \times \mathbf{x}$; $\mathbf{w} \in \mathbb{Z}^{30 \times 30}$, $\mathbf{x} \in \mathbb{Z}^{30}$) by reducing the computational complexity to $O(n)$. To this end, we aimed to calculate a dot product $\mathbf{w}[i, :] \cdot \mathbf{x}$ at one cycle, where $\mathbf{w}[i, :]$ denotes the i th row of matrix \mathbf{w} . We restricted the elements w of matrix \mathbf{w} to 2-bit integers ($w \in \{0, 1, 2, 3\}$) and the elements x to 1-bit integers ($x \in \{0, 1\}$). The matrix \mathbf{w} was transposed and mapped onto our 30×30 SRMC CA (conductance of each cell $\in \{HRS, L1, L2, L3\}$). The vector \mathbf{x} with 1-bit integer elements was encoded as a voltage array V_{ap} ($V_{ap} \in \{0, 2V\}$) and applied to the 30 row-lines of the CA (Fig. 8a). The current measured at the i th column-line was the intermediate result of the dot product $\mathbf{w}[i, :] \cdot \mathbf{x}$. As depicted in Fig. 8b, we addressed one column at one cycle by pulling down the chosen column-line to the ground while inhibit voltages ($V_{inhibit}$) were applied to the rest of column-lines ($2/3V_{ap}$), so that we reduced the complexity to $O(n)$, which is otherwise $O(n^2)$.

We chose four random matrices (\mathbf{w}_1 , \mathbf{w}_2 , \mathbf{w}_3 , and \mathbf{w}_4) of different sparsities (0, 25, 51, and 55%, respectively). The percentage of each integer (0, 1, 2, 3) in each matrix is shown in Fig. 8c. The chosen matrices were mapped onto four 30×30 CAs such that the individual cells of the CAs were randomly accessed and programmed to the correct conductance states using Scheme 2. The programmed conductance map for each matrix is shown in Fig. 8d – g. The conductance of each SRMC was individually read out at a read-out voltage of 2 V to acquire the maps. We then

performed the dot product $\mathbf{w}[i, :] \cdot \mathbf{x}$ for each i at one cycle with vector \mathbf{x} of ones, i.e., $\mathbf{x} = [1, 1, \dots, 1]$. The vector-matrix multiplication operation for each matrix thus consumes 30 column-line-addressing cycles, yielding a current vector \mathbf{j} ($\in \mathbb{R}^{30}$) as the intermediate product (Fig. 8d – g). The measured current at each column-line is almost identical to the current value extrapolated from each cell current in the same column, indicating marginal disturbance from the unselected cells. For the multiplication with four matrices ($\mathbf{w}_1, \mathbf{w}_2, \mathbf{w}_3$, and \mathbf{w}_4), the CA domain consumes powers of 4.22, 3.44, 2.83, and 2.69 \square W, respectively. The considered multiplication is the worst case in terms of power consumption because of the extremely dense vector \mathbf{x} (of ones).

To output the final product \mathbf{z} ($\mathbf{z} = \mathbf{w} \times \mathbf{x}$; $\mathbf{z} \in \mathbb{Z}^{30}$), current from the i th column j_i for all i needs to be encoded as a binary number, which subsequently enters into the near-memory digital domain for additional processing. A common method is to convert the summed current to voltage and subsequently to quantize the converted voltage using an analog-to-digital converter (ADC)⁵⁰. Alternatively, the summed current can directly be converted to a binary value using a current sense amplifier (CSA) with multiple reference currents that are iteratively compared with the summed current⁵¹. In either way, the important consideration is twofold: (i) energy consumption and (ii) bit-width of the product \mathbf{z} . Regarding power consumption, ADCs are well known to consume a considerable amount of energy inasmuch as the total energy consumption of an RRAM-based inference accelerator is dominated by the ADCs¹³. An alternative method using a CSA⁵¹ keeps the static current from the chosen line flowing while the summed current being converted iteratively, causing additional energy consumption. The bit-width should be chosen carefully to avoid the performance, i.e., inference, degradation by the quantization bit-width. As shown in quantized neural networks such as DoReFa-Net⁵², the resolution, i.e., bit-width, of activations more critically determines the inference accuracy than that of weights. The activation resolution is dictated by the bit-width of the output \mathbf{z} . Therefore, the bit-width of the product \mathbf{z} is an important consideration in the design of summed current-encoding circuits.

Regarding multibit factor \mathbf{x} , time-division multiplexing is a desirable method by encoding the vector \mathbf{x} as shown in Fig. 9. Because of the nonlinear I - V behavior in the LRS of our SRMCs, encoding a factor as input voltage amplitude is unsuitable unlike linear I - V cases^{14, 53}. The l -bit elements $x[i]$ are time-division multiplexed from the least significant bits (LSBs) to the most significant bits (MSBs) and are applied to the row-lines at one column-line addressing cycle for the dot product $\mathbf{w}[i, :] \cdot \mathbf{x}$. Thus, each dot product cycle includes l sub-cycles. The output current at each sub-cycle is encoded as a binary value and subsequently multiplied by 2^{k-1} , where k denotes the digit corresponding to the sub-cycle. The results are finally summed to output the dot product $\mathbf{w}[i, :] \cdot \mathbf{x}$.”

Reviewers' Comments:

Reviewer #2:

Remarks to the Author:

The authors have performed additional modeling and experiments. The manuscript is now significantly improved. However, since the 100kb array is not functional, the title needs to be changed. For example, removing "100kb" in the title may work. I think the manuscript can be published after addressing this concern.

Reviewer #3:

Remarks to the Author:

I thank the authors for their replies. I am fine with them, I just have one final question. Why are you forced to implement matrix-vector multiplication one column each time, leading to $O(n)$ complexity? What is preventing from performing $O(1)$ complexity, in other words activating all rows and all columns at the same time? This is the main advantage for crossbar arrays, so this property would be highly desirable.

Responses to reviewers' comments

Manuscript #: NCOMMS-20-27954A

Title: Self-rectifying resistive memory in 100 kb passive crossbar arrays

Authors: Kanghyeok Jeon, Jeeson Kim, Jin Joo Ryu, Seung-Jong Yoo, Choongseok Song, Min Kyu Yang, Doo Seok Jeong, Gun Hwan Kim

We greatly acknowledge the valuable comments on our manuscript from the reviewers. Based on the reviewers' comments and suggestions, we revised our manuscript thoroughly. The revision made is highlighted using blue color.

Reviewer #2 (Remarks to the Author):

The authors have performed additional modeling and experiments. The manuscript is now significantly improved. However, since the 100kb array is not functional, the title needs to be changed. For example, removing "100kb" in the title may work. I think the manuscript can be published after addressing this concern.

Answer: We agree on the suggestion.

Author action: We changed the title to "Self-rectifying resistive memory in passive crossbar arrays".

Reviewer #3 (Remarks to the Author):

I thank the authors for their replies. I am fine with them, I just have one final question. Why are you forced to implement matrix-vector multiplication one column each time, leading to $O(n)$ complexity? What is preventing from performing $O(1)$ complexity, in other words activating all rows and all columns at the same time? This is the main advantage for crossbar arrays, so this property would be highly desirable.

Answer: We definitely understand the reviewer's concern. Crossbar arrays may achieve the ultimate complexity $O(1)$ only if non-ideal factors such as sneak current and line resistance effect are excluded. Fortunately, the sneak current effect may be ruled out even if all column lines are simultaneously addressed, i.e., grounded, because all bit-cells are non-negatively biased in this case. However, the effect

of finite line resistance remains significant. The finite line resistance causes the inhomogeneous distribution of bit-cell voltages over the cells on the same row line such that the further a bit-cell from the row line contact, the lower voltage is applied across the bit-cell. Further, this effect is boosted when the bit-cells on the same row line simultaneously allow current flow, which is the case of column lines that are simultaneously addressed. Despite the high resistance level of our SRMC in both states the line resistance effect comes into play if the crossbar array is sufficiently large to allow considerable amount current through the parallel bit-cells on the same row line.

Additionally, simultaneously addressing all column lines requires a current sense amplifier (CSA) and following logic circuit per column line, resulting in a considerable area overhead. The sequence address scheme allows one CSA to be shared among a group of column lines through time-division multiplexing because the CSA is addressed by one column line at any one cycle. Therefore, we find the reduction in complexity to $O(n)$ practical.

Reviewers' Comments:

Reviewer #3:

Remarks to the Author:

Dear authors, thanks for your reply. In essence it looks to me that the choice of performing computation one column each time looks more related to practical issues (line resistance, sense amplifier size overhead), which, in a future fab design at a scaled node, would eventually be solved.

I would just add in the text a short sentence mentioning such aspect, and then the paper is fine for me, no need to resubmit for review.

Responses to reviewers' comments

Manuscript #: NCOMMS-20-27954B

Title: Self-rectifying resistive memory in passive crossbar arrays

Authors: Kanghyeok Jeon, Jeeson Kim, Jin Joo Ryu, Seung-Jong Yoo, Choongseok Song, Min Kyu Yang, Doo Seok Jeong, Gun Hwan Kim

We greatly acknowledge the valuable comment on our manuscript from the reviewer. Based on the reviewer's comment and suggestion, we revised our manuscript thoroughly. The revision made is highlighted using blue color.

Reviewer #3 (Remarks to the Author):

Dear authors, thanks for your reply. In essence it looks to me that the choice of performing computation one column each time looks more related to practical issues (line resistance, sense amplifier size overhead), which, in a future fab design at a scaled node, would eventually be solved.

I would just add in the text a short sentence mentioning such aspect, and then the paper is fine for me, no need to resubmit for review.

Answer: Thank the reviewer for the comment.

Author action: We added the related part in the revised manuscript like below.

Ideally, CAs may achieve the ultimate complexity $O(1)$ of vector-matrix multiplication beyond the complexity $O(n)$ by addressing all column-lines at one cycle. The basic premise is that all non-ideal factors, e.g., sneak current and line resistance effects, are excluded. The sneak current effect may be marginal because all bit-cells are supposed to be non-negatively biased when all column-lines are simultaneously addressed, i.e., grounded. However, the effect of finite line resistance is significant. The finite line resistance causes the

inhomogeneous distribution of bit-cell voltages over the cells on the same row-line such that the further a bit-cell from the row-line contact, the lower voltage is applied across the bit-cell. Further, this effect is boosted when the bit-cells on the same row-line allow simultaneous current flow, which is the case of all column-line addressing.

Additionally, simultaneously addressing all column-lines requires one CSA and following logic circuit per column line, whereas addressing one column-line at a cycle allows one CSA to be shared among a group of column-lines through time-division multiplexing. This additional peripheral circuit-area overhead can be prohibitive in large-scale CAs. Therefore, the complexity reduction to $O(1)$ may be realized only when these challenges are overcome.